# Gating mechanism of elongating β-ketoacyl-ACP synthases

Jeffrey T. Mindrebo[1,2,5], Ashay Patel[1,5], Woojoo E. Kim[1], Tony D. Davis [1], Aochiu Chen [1], Thomas G. Bartholow[1], James J. La Clair[1,2], J. Andrew McCammon[1,3], Joseph P. Noel [1,2,4 ✉] & Michael D. Burkart [1 ✉]

Carbon-carbon bond forming reactions are essential transformations in natural product biosynthesis. During de novo fatty acid and polyketide biosynthesis, β-ketoacyl-acyl carrier protein (ACP) synthases (KS), catalyze this process via a decarboxylative Claisen-like condensation reaction. KSs must recognize multiple chemically distinct ACPs and choreograph a ping-pong mechanism, often in an iterative fashion. Here, we report crystal structures of substrate mimetic bearing ACPs in complex with the elongating KSs from *Escherichia coli*, FabF and FabB, in order to better understand the stereochemical features governing substrate discrimination by KSs. Complemented by molecular dynamics (MD) simulations and mutagenesis studies, these structures reveal conformational states accessed during KS catalysis. These data taken together support a gating mechanism that regulates acyl-ACP binding and substrate delivery to the KS active site. Two active site loops undergo large conformational excursions during this dynamic gating mechanism and are likely evolutionarily conserved features in elongating KSs.

[1] Department of Chemistry and Biochemistry, University of California, San Diego, 9500 Gilman Drive, La Jolla, CA 92093-0358, USA. [2] Jack H. Skirball Center for Chemical Biology and Proteomics, Salk Institute for Biological Studies, La Jolla, CA 92037, USA. [3] Department of Pharmacology, University of California, San Diego, 9500 Gilman Drive, La Jolla, CA 92093, USA. [4] Howard Hughes Medical Institute, Salk Institute for Biological Studies, La Jolla, CA 92037, USA. [5]These authors contributed equally: Jeffrey T. Mindrebo, Ashay Patel ✉email: noel@salk.edu; mburkart@ucsd.edu

Fatty acid synthases (FASs) and polyketide synthases (PKSs) iteratively condense and modify ketide units to produce a variety of natural compounds, ranging from fatty acids to complex bioactive molecules[1,2]. FASs and PKSs can exist as one or more polypeptide mega-synthases that contain distinct catalytic domains (type I) or as discrete enzymes, each possessing specific activities (type II)[3–8]. In these pathways, β-ketoacyl ACP synthases, alternatively ketosynthases (KSs), catalyze carbon–carbon formation via a precisely choreographed decarboxylative Claisen-like condensation to produce a β-ketoacyl species. In fatty acid biosynthesis (FAB, Fig. 1a), complete reduction at the β-position occurs before further chain extension, while in polyketide biosynthesis (PKB), the β-positions of pathway intermediates can be completely, partially, or not reduced to produce structurally and functionally complex natural products[1,9]. The small, 4-α-helical acyl carrier protein (ACP) is used to shuttle substrates and intermediates to each catalytic domain or discrete enzyme in these pathways[10,11]. ACPs are initially translated to an inactive *apo* form and are subsequently post-translationally modified to an active *holo* form via attachment of a 4′-phosphopantetheine (PPant) arm at a conserved serine residue, providing a thiol moiety that ligates substrates and pathway intermediates to the ACP[10]. Protein–protein interactions (PPIs) between ACPs and their enzymatic partners (or domains) regulate the catalytic activities of FASs and PKSs and additionally provide a mechanism to ensure pathway orthogonality between primary (i.e., FAB) and secondary metabolism (i.e., PKB)[3,5,6,12–17].

KS-catalyzed carbon–carbon bond formation is an essential reaction common to both FAB and PKB. The decarboxylative condensation reaction provides a thermodynamic driving force for the formation of complex natural products[18,19]. In addition, KS domains play an important role in chain length determination,

ensuring product fidelity, and controlling pathway flux[15,20,21]. Therefore, efforts to engineer FASs and PKSs will depend on our ability to manipulate the specificities and activities of these domains/enzymes, as evidenced by the recent successes engineering the yeast FAS[21–26].

KSs can be classified as elongating or initiating based on their role within a given pathway. Initiating KSs serve to catalyze the first condensation reaction of FAB or PKB pathways, converting *holo*-ACP to acyl-ACP, while all subsequent condensation reactions are catalyzed by elongating KSs[3,4]. Despite their distinct biosynthetic roles, both types of KS operate via a common ping–pong reaction mechanism, which can be disaggregated into two discrete steps[21,27,28]. In the first step, an acyl-ACP associates with a KS, and its acyl cargo is transferred from the PPant arm to a conserved, active site cysteine, producing a covalent acyl-KS adduct and *holo*-ACP. In the second step, malonyl-ACP replaces *holo*-ACP as the KS's binding partner, subsequently undergoing a decarboxylative Claisen-like condensation with the acyl-KS adduct that produces β-ketoacyl-ACP (Supplementary Fig. 1). During the course of this ping-pong process, two distinct ACPs must sequentially interact with the KS, and thus must enforce exquisite temporal and spatial discrimination among the subtly different chemical states of these acyl-ACPs. The mechanisms used by KSs to coordinate these two half-reactions and how this coordination determines substrate specificity remains cryptic despite extensive biochemical and structural characterization.

The *E. coli* type II FAS has served as a model system for understanding ACP-mediated PPIs and FAB[19,29–36]. Here, we use this well-characterized system to better understand ACP·KS PPIs and KS substrate discrimination by structurally characterizing *E. coli* elongating KSs, FabB (KASI family KS), and FabF (KASII

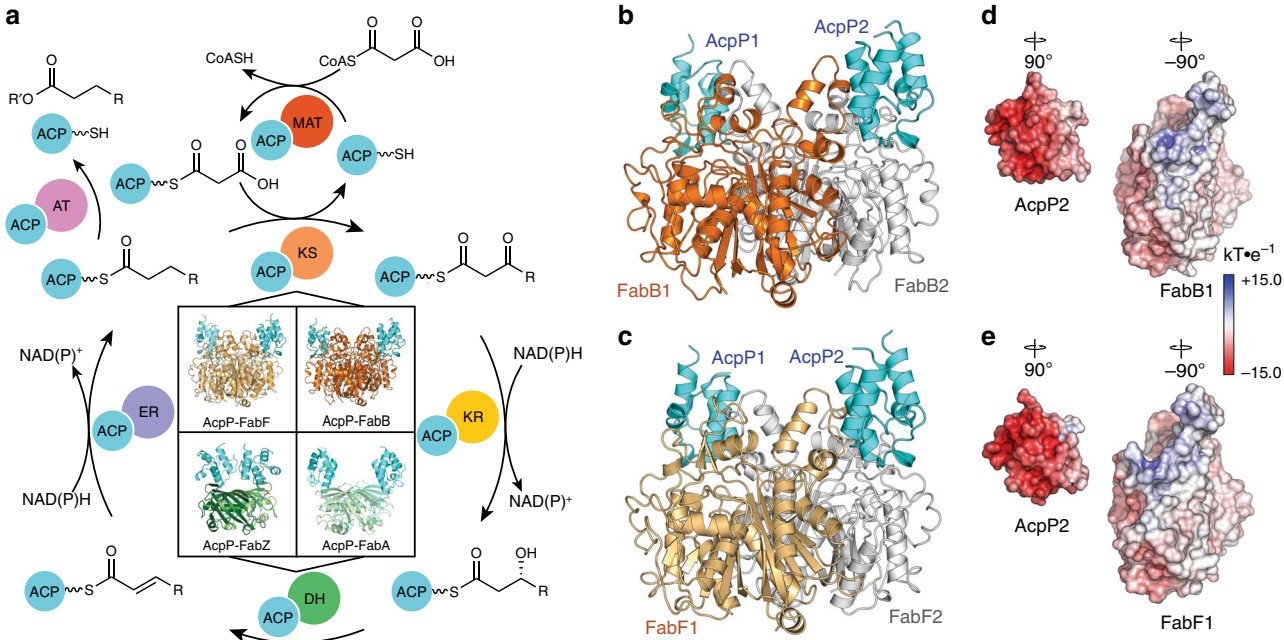

**Fig. 1 FAB cycle and known AcpP–PP complexes. a** A schematic representation of the chemical transformations that occur during fatty acid biosynthesis, highlighting the importance of protein–protein interactions required at each step for substrate processing. The four quadrants at the center of the scheme show the structures of all FAS AcpP crosslinked complexes solved thus far. The top two structures are the crosslinked AcpP–KSs (AcpP–FabF and AcpP–FabB) reported herein, while depicted below are the crosslinked AcpP–DHs (AcpP–FabZ and AcpP–FabA) solved and reported in previous publications[14,33]. Each domain is abbreviated as follows: MAT malonyl-ACP transacylase, KS ketosynthase, KR ketoreductase, DH dehydratase, ER enoylreductase, and AT acyl transferase. AcpP-DH structures depicted in bottom two quadrants are AcpP-FabZ (PDB: 6N3P), AcpP-FabA (PDB: 4KEH). **b, c** Cartoon representations of the crosslinked AcpP–FabB and AcpP–FabF dimers. AcpP is colored cyan; the first KS monomer of AcpP–FabB and AcpP–FabF is colored orange or light orange, respectively, while the second is colored white in both structures. **d** Electrostatic potentials (ESP) maps of the FabB and AcpP monomers. **e** ESP maps of the FabF and AcpP monomers. In all cases, the ESP is mapped onto the Connelly surfaces of the FabF and AcpP monomers using a blue to white to red color range, spanning from +15.0 kT·e⁻¹ to –15.0 kT·e⁻¹.

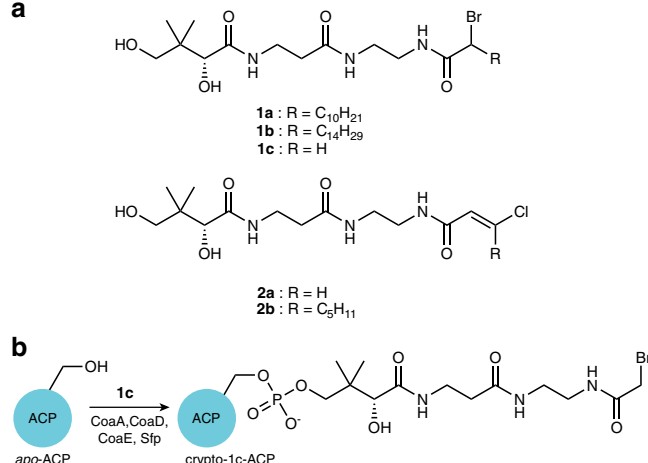

**Fig. 2 KS pantetheineamide crosslinking probes and ACP one-pot method. a** Pantetheineamide-derived α-bromo and chlorovinyl crosslinking probes used in this study. **b** An illustration of the one-pot chemoenzymatic loading of apo-ACP with panetheinamide probe **1c** to produce crypto-**1c**-ACP using the coenzyme A biosynthetic enzymes CoaA, CoaD, and CoaE and a promiscuous phosphopantetheinyl transferase (PPTase) Sfp.

family KS), as substrate/intermediate complexes with ACP that approximate states formed during catalysis. In conjunction with molecular dynamics (MD) simulations and mutagenesis studies, these structures provide dynamical descriptions of two ACP·KS complexes, revealing conformational changes that regulate substrate processing. Specifically, we propose that the elongating KSs employ two conserved active site loops that open and close through a double drawbridge-like gating mechanism in order to regulate substrate processing[37]. When open, the drawbridge expands the KS active site in order to accommodate PPant-tethered acyl-AcpP substrates. The structural features underlying this gating machinery are conserved in related enzymes and similar conformational transitions will likely determine substrate selectivity and processing in other FAB and PKB pathways.

## Results

**Crosslinking and crystallization of AcpP–KS complexes.** We synthesized C12-α-bromo- (**1a**) and C16-α-bromopantetheine-amide (**1b**) and loaded them onto the *E. coli* ACP (AcpP) using a one-pot chemoenzymatic method[38] to produce crypto-**1a**-AcpP (C12AcpP) and crypto-**1b**-AcpP (C16AcpP) that bear a reactive warhead on their respective PPant arms[38–40] (Fig. 2). These crypto-AcpPs were then reacted with FabF and FabB to produce uniformly crosslinked complexes, referred to herein as AcpP–KS, where "-" denotes a crosslinked complex (Supplementary Figs. 1 and 2). Using this method, we grew crystals of C12AcpP–FabF and C16AcpP–FabF and obtained x-ray crystallographic data sets with 2.35 Å and 2.30 Å resolution, respectively. The crystallographic asymmetric units (ASU) of these complexes were identical and contained one monomer of FabF crosslinked to a single AcpP. In addition, C12AcpP–FabB crystals diffracted to 1.55 Å and C16AcpP–FabB diffracted to 2.50 Å. The C12AcpP–FabB and C16AcpP–FabB ASUs both contain the functional FabB dimer crosslinked to two AcpP molecules (Fig. 2b, c). Additional analysis for the C12AcpP–FabF and C16AcpP–FabB structures can be found in Supplementary Notes 1 and 2.

**The AcpP–KS interface.** The AcpP–KS interacting surfaces are electrostatic (Fig. 2b–e) and composed largely of flanking charged and polar groups surrounding a central hydrophobic patch (Fig. 3). The AcpP and KS monomers provide negatively and positively charged residues, respectively, to the interface. Figure 3 provides a comparison of the AcpP–FabF and AcpP–FabB interfaces from this work. While a thorough analysis of the AcpP–FabB interface was recently performed by Milligan et al.[34], herein we report an example of a KASII-type KS in complex with an ACP, AcpP–FabF. The AcpP–FabF and AcpP–FabB complexes can be broken into three regions. Region 1 electrostatic interactions include Lys65, Arg68, and Lys69 of FabF interacting with Glu13, Gln14, Asp35, and Asp38 of AcpP (Fig. 3a). In addition, Thr270 of FabF, which sits on a flexible loop that undergoes conformational rearrangements upon acyl-AcpP binding, interacts with Asp35 of AcpP. Region 2 interactions are formed between Arg127, Lys128, and Arg206 of FabF and Glu47, Glu48, Glu53, and Asp56 of AcpP (Fig. 3b). The central hydrophobic patch is comprised of interactions between Met44, Val40, and Leu37 of AcpP and Ile129, Ser130, Pro131, Phe132, Ala205, and Arg206 of FabF (Fig. 3c).

Comparisons of the AcpP–FabF and AcpP–FabB interfaces reveal that they share similar chemical features, but structural alignments of the KS subunits show that the orientations of AcpPs in the AcpP–KS complexes are distinct (Supplementary Fig. 3a, b). In FabF, individual ACP monomers, AcpP1 and AcpP2, bury surface areas of 671.2 Å$^2$ and 675.4 Å$^2$, respectively, while in FabB, AcpP1, and AcpP2 bury surface areas of 597.5 Å$^2$ and 607.4 Å$^2$, respectively. These small contact areas are consistent with the transient nature of molecular recognition between ACP and its binding partners[14,33]. In addition, the differences in contact areas of AcpP–FabF and AcpP–FabB interfaces are consistent with the differing orientations that the AcpPs adopt relative to their KS partners. It is also worth noting that when comparing the interfaces of these AcpP–KS complexes, AcpP's helix I forms more extensive contacts with FabF than FabB, while FabB forms more hydrophobic interactions at region 3 (Fig. 3). While the differences at these interfaces are subtle, the manner in which AcpP engages with each of its cognate KSs may modulate the activities of FabF and FabB[41,42]. A time-resolved analysis of the FabF and FabB interfaces using MD simulation data is provided in Supplementary Note 3.

**Changes in the FabF active site accompany substrate binding.** Electron density maps calculated using the 2.30 Å C16AcpP–FabF data set demonstrate adequate density for ACP's Ser36, the PPant moiety, and acyl cargo. The FabF active site cysteine residue (Cys163) forms a covalent bond with the α-carbon of the fatty acid substrate mimetic (**1b**), and the resulting stereocenter is found in the S configuration (Fig. 4a). The carbonyl group of the substrate coordinates to the FabF active site histidine residues, His303 and His340, in a manner similar to the published structure of AcpP–FabB[34] (PDB: 5KOF) (Fig. 4b). Modeling and refinement of the crosslinker's acyl chain required the remodeling of two active site loops. The first of which is the conserved GFGG loop (loop 1, residues 399–402) adjacent to the active site that contains the putative Phe400 gating residue[27], and a second less conserved loop (loop 2, residues 265–275) that caps loop 1 (Fig. 5a). These loops undergo significant conformational changes, revealing a hydrophobic pocket that accommodates the acyl chain (Supplementary Fig. 4a–c and Supplementary Note 4). This pocket is distinct from the substrate binding site identified in the crystallized acyl-KS intermediate structures 2GFY[43] and 1EK4[44] (Fig. 4c). Close analysis of the C16AcpP–FabF structure indicates that crystal contacts do not play a role in the conformational changes seen in these loops.

**C16AcpP–FabF adopts an open conformation.** A comparison of the *apo*-FabF (PDB: 2GFW)[43], acyl-FabF (PDB: 2GFY)[43],

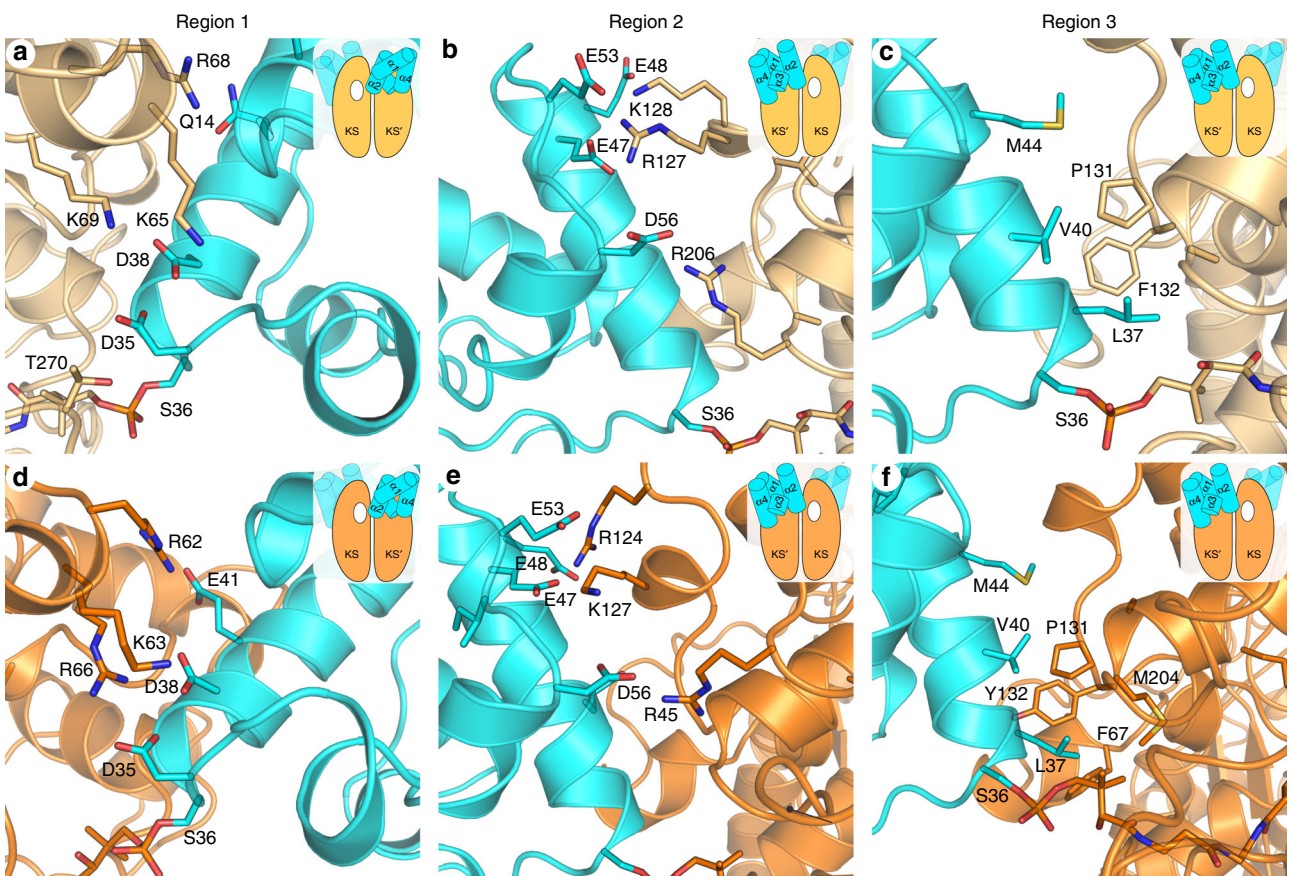

**Fig. 3 AcpP–FabF and AcpP–FabB interface interactions.** The FabF (light orange) and FabB (bright orange) interfaces can be broken up into three regions, the first two are comprised of electrostatic interactions and the third is a hydrophobic patch. The above panels provide close-up views of these three regions and the cartoon inset provides context regarding the viewing angle and the positional relationship of AcpP (cyan) to the overall AcpP–KS complex. In addition, the white hole represents the active site to which the viewed AcpP is crosslinked as the majority of contacts are provided by the KS monomer not covalently crosslinked to AcpP. **a** A close-up view of the AcpP–FabF region 1 electrostatic interactions. **b** A close-up view of the AcpP–FabF region 2 electrostatic interactions. **c** A close-up view of the AcpP–FabF region 3 hydrophobic interactions. **d** A close-up view of the AcpP–FabB region 1 electrostatic interactions. **e** A close-up view of the AcpP–FabB region 2 electrostatic interactions. **f** A close-up view of the AcpP–FabB region 3 hydrophobic interactions.

and C16AcpP–FabF structures shows that Phe400 of loop 1 moves ~7 Å, as measured by changes in the Phe400 Cα position, into a pocket revealed by a conformational rearrangement of loop 2 (Fig. 5a). Gly399 and Gly402 of loop 1 act as pivot points as the loop transitions from a closed conformation to an open conformation (Fig. 5b, c). The open conformation is stabilized by a network of hydrogen-bonding interactions between the backbone amides of Gly399, Phe400, Gly401, and Gly402 and the side chains of Asn404 and Asp265 from loop 2 (Fig. 5b, c). Overlays of the *apo*-FabF and C16AcpP–FabF structures indicate that the C16-acyl chain of **1b** would clash with Phe400 in the closed conformation of loop 1 (Fig. 5a). These results indicate that these loop rearrangements may act as a gate to provide access to the active site for acyl-ACP substrates. Interestingly, the backbone amide of Phe400 provides one of the two hydrogen bonds that form the FabF oxyanion hole when loop 1 assumes the closed conformation; therefore, the open conformation is likely catalytically inactive.

In *apo*-FabF, loop 2 rests on top of loop 1 and must undergo a rearrangement to allow loop 1 to assume the open conformation. A comparison of *apo*-FabF to our C16AcpP–FabF structure demonstrates that loop 2 pivots about a Pro–Pro motif, whereby Pro273 undergoes a 165° change in its ψ angle, causing the loop to swing outwards by 5 Å towards the bound AcpP (Fig. 5d). Analysis of the loop 2 backbone dihedral angles suggests that the loop transitions from a type VIII β-turn in the

closed conformation to a type I β-turn in the open conformation. In the closed conformation, loop 2 is stabilized by a $Ser271_{(O)C}$-$His268_{(H)N}$ main-chain hydrogen bond, a Thr270-His268 side-chain hydrogen bond, and a Pro273-Tyr267 C–H π interaction[45] (Fig. 5e, f). Transition to the open conformation causes the β-turn backbone hydrogen-bonding pattern to flip from $Ser271_{(O)C}$-$His268_{(H)N}$ to $Ser271_{(N)H}$-$His268_{(O)C}$ and Thr270 breaks its hydrogen bond with His268 to interact with Asp35 of AcpP.

**C12AcpP–FabB trapped in closed acyl transfer conformation.** Analysis of the 1.55 Å C12AcpP–FabB data (Supplementary Fig. 4d) demonstrates well-resolved electron density for the 4′-phosphopantetheineamide arm and the C12-acyl chain. In this structure, loops 1 and 2 are found in a catalytically competent closed conformation (Fig. 4d–f). Like the catalytic cysteine of FabF, Cys163 of FabB, reacts stereoselectively with the α carbon of **1a** to produce the *S* stereoisomer (Fig. 4d). The carbonyl group of the substrate is coordinated to the oxyanion hole formed by the backbone amides of Cys163 and Phe400, and the probe's acyl chain is positioned inside FabB's fatty acid binding pocket (Supplementary Fig. 4d, e). Interestingly, the recently deposited AcpP–FabB structure (PDB: 5KOF)[34] does not feature these interactions, and instead the carbonyl group forms hydrogen bonds with His298 and His333.

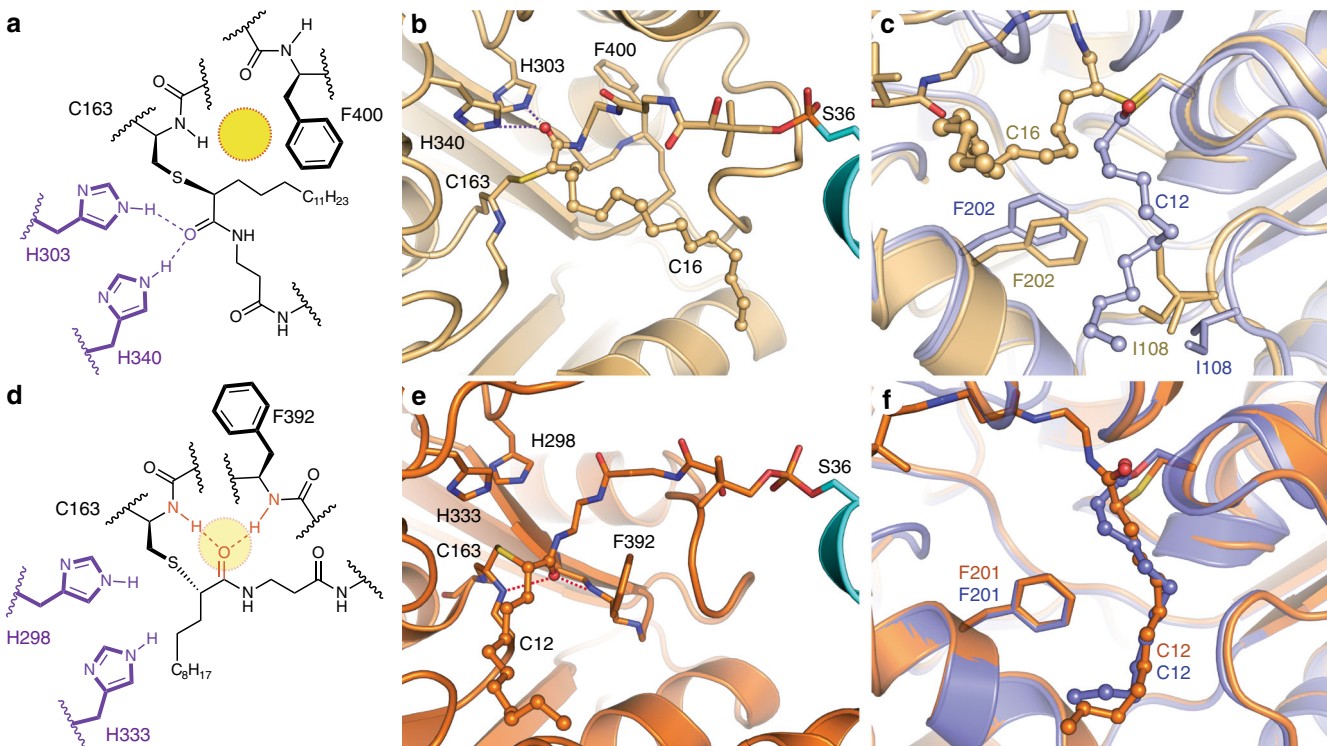

**Fig. 4 FabF active site interactions with C16 substrate analog. a** Schematic representation of the 4′-phosphopantetheineamide crosslinker within the FabF active site. The disorganized oxyanion hole is shown as a yellow circle. **b** Active site interactions depicting the PPant crosslinker and C16 acyl chain from the crystal structure of C16AcpP–FabF. H340 and H303 coordinate the carbonyl of the acyl mimetic (blue dotted lines) and the active site oxyanion hole is disorganized. AcpP and FabF are colored cyan and light orange, respectively. **c** Comparison of acyl binding pocket and associated gating residues, F202 and I108, in dodecanoyl-FabF (12:0-FabF) (2GFY[43], light purple), and C16AcpP–FabF (6OKG, light orange). **d** Schematic representation of the 4′-phosphopantetheine crosslinker within the FabB active site. The organized oxyanion hole is shown as a transparent yellow circle with associated backbone amides and coordinated PPant carbonyl shown in red. **e** The carbonyl of the acyl mimetic in the crystal structure of C12AcpP–FabB is coordinated in the active site oxyanion hole formed by backbone amides of Cys163 and Phe392 (Phe400 in FabF). **f** Comparison of acyl binding pocket in dodecanoyl-FabB (12:0-FabB) (PDB ID: 1EK4[44], bright purple), and C12AcpP–FabB (PDB ID: 6OKC, bright orange). Residues in panels are numbered according to their respective protein, FabF is numbered in FabF residue numbering and FabB is numbered in FabB residue numbering. Hydrogen-bonding interactions are represented as dotted lines, all side chain residues are represented as sticks, and the 4′-phosphopantetheinyl arm is represented as sticks while the acyl chains are represented as ball and stick.

Comparison of C12AcpP–FabB and dodecanoyl-FabB (C12-FabB, PDB: 1EK4)[44] demonstrates that the active site and acyl chain geometries are nearly identical in the two structures (Fig. 4f). The similarity between the C12-FabB and C12AcpP–FabB indicates that the latter approximates a catalytically relevant state for the transfer of substrate from acyl-AcpP to FabB's active site cysteine. Both steps of the KS reaction form a tetrahedral intermediate that is stabilized by the oxyanion hole, which is only organized when loop 1 and the associated Phe392 residue (Phe400 in FabF), are in the gate-closed conformation.

**MSA analysis suggests conservation of loops 1 and 2.** Multiple sequence alignments (MSA) of type II FAS elongating KSs indicate that residues of loop 1 are highly conserved across the KASI (FabB) and KASII (FabF) families of KSs, while loop 2 is only conserved within, but not between, the two families (Supplementary Fig. 5). The consensus sequence for loop 1 is comprised of the conserved GFGG β-turn motif flanked by two conserved asparagine residues, Asn396 and Asn404 (FabF numbering). Interestingly, Asn404 interacts with a highly conserved Asp265 found at the start of loop 2 in both the open and closed states, providing a physical basis for the (proposed) coordinated movement of loops 1 and 2 (Fig. 5b, c). The conserved nature of loop 1 and Asp265 suggests that their proposed gating function is

a general feature of type II FAS elongating KSs. Further analysis and comparison of FabF and FabB structure and function are provided in Supplementary Note 5.

To determine if these proposed gating elements are conserved in condensing enzymes outside of type II FAS, we aligned *E. coli* FabF with representative KSs from type II PKS, type I FAS, and type I PKS (Fig. 6a). Loop 1 of FAS and PKS KSs is generally conserved, although there is modest sequence variation. Notably, the Phe400 FabF gating residue is conserved in type I FAS and type II PKS but is replaced by valine or isoleucine in the KS domains of the type I PKS deoxyerythronolide B synthase (DEBS). Modification of the gating residue may modulate substrate specificity, as the KS domains from DEBS accept and extend substrates with β-substituents and utilize bulkier alpha-branched extender units, namely methylmalonyl-CoA[46].

Loop 2 of these KSs is significantly less conserved than loop 1. Only Asp265 can be considered highly conserved, as it is present in all KSs except the KS monomer (actKS) of the heterodimeric type II PKS KS/chain length factor (CLF) complex from actinorhodin biosynthesis[47,48]. In contrast to FAS KSs, these enzymes iteratively extend a growing polyketide within the KS binding pocket only unbinding the polyketide once multiple extensions using malonyl-ACP have been catalyzed to produce a fully extended poly-β-keto-ACP. The size of the polyketide produced by actKS is determined by CLF, which accommodates

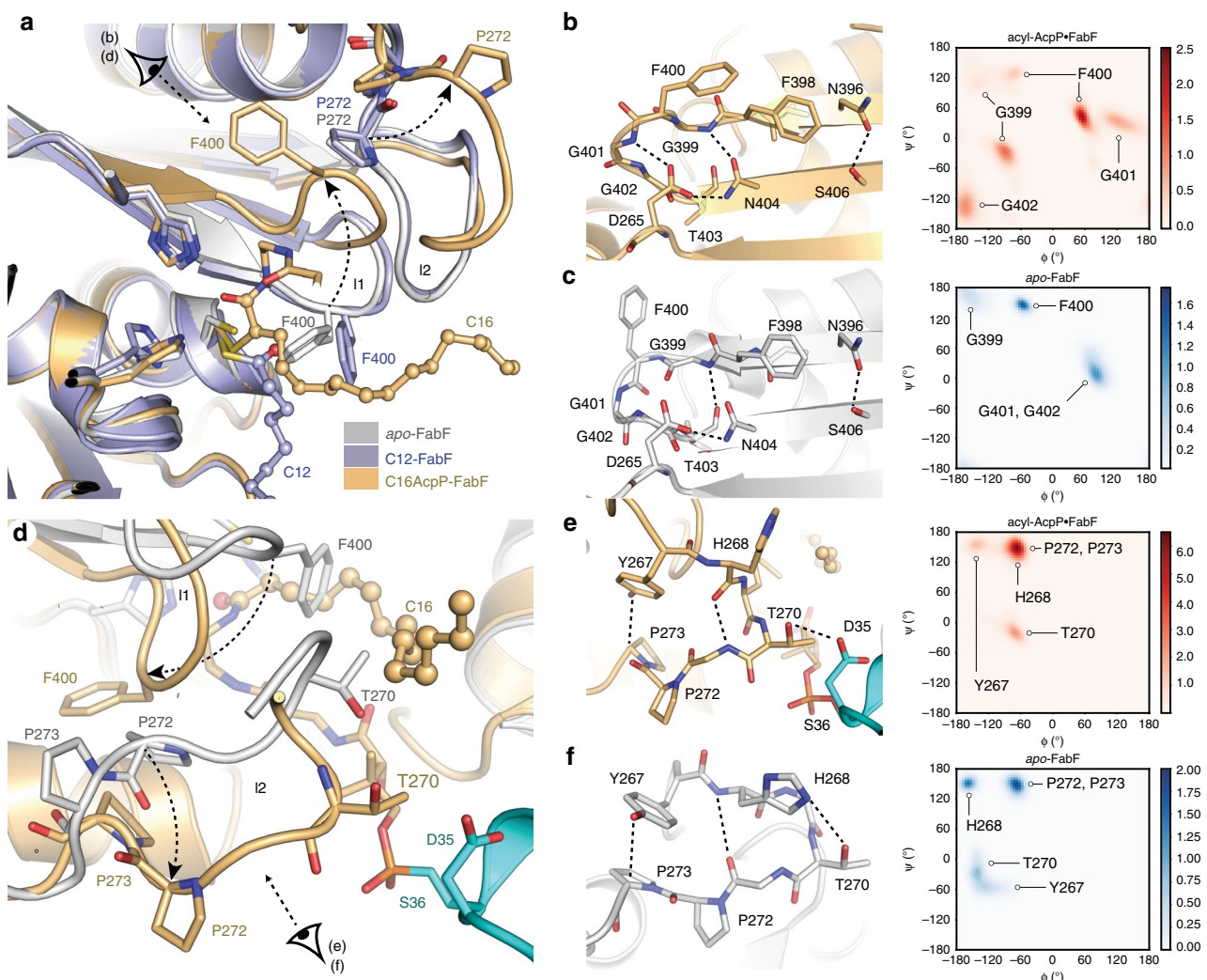

**Fig. 5 Conformational changes in FabF active site loops. a** Comparison of F400, loop 1, and loop 2 conformations of *apo*-FabF (light grey, PDB: 2GFW[43]), dodecanoyl-FabF (light purple, PDB: 2GFY[43]), and C16AcpP–FabF (light orange, PDB: 6OKG). **b** Interactions that stabilize the open-conformation of the β-turn motif of loop 1 in C16AcpP–FabF and associated Ramachandran analysis of the conserved GFGG sequence within loop 1 from 1.5 μs of 10:0-AcpP·FabF, 12:0-AcpP·FabF, 16:0-AcpP·FabF (total of 4.5 μs) molecular dynamics (MD) simulation data. **c** Interactions that stabilize the closed-conformation of the β-turn motif of loop 1 in *apo*-FabF (light grey) and associated Ramachandran analysis of the conserved GFGG sequence within loop 1 from 1.5 μs of MD simulation data of *apo*-FabF. **d** Loop 2 overlay of *apo*-FabF (light grey, PDB: 2GFW) and C16AcpP–FabF (light orange) with the crosslinked AcpP colored cyan. **e** Interactions that stabilize the β-turn open conformation of loop 2 in C16AcpP–FabF (light orange) and associated Ramachandran analysis of key loop 2 residues of FabF monomers of acyl-AcpP·FabF complexes (gate-open conformation) from 1.5 μs of 10:0-AcpP·FabF, 12:0-AcpP·FabF, 16:0-AcpP·FabF (total of 4.5 μs) MD simulation data. **f** Interactions that stabilize the loop 2 closed conformation in *apo*-FabF (white) and associated Ramachandran analysis of key loop 2 residues of FabF of *apo*-FabF (gate-closed conformation) from 1.5 μs of MD simulation data of *apo*-FabF. **a–f** Residues are shown as sticks and colored according to element. Hydrogen-bond interactions are highlighted using dotted lines. **b**, **c**, **e**, **f** Simulated backbone φ and ψ dihedrals were binned using widths of 5°. Color bars indicates density of data points within each 2D bin.

the growing polyketide chain, mediating substrate dissociation when the polyketide intermediate can no longer be accommodated within its binding pocket[49]. Therefore, given that iterative type II KSs do not accept acyl-ACPs and instead only process malonyl-ACPs, these iterative type II KSs may need to disfavor gating events to prevent premature product off-loading and ensure the growing polyketide is transferred back to the KS cysteine after condensation with malonyl-ACP. In actKS, this may be realized by disfavoring the open conformation of loops 1 and 2, relative to type II FAS KSs, through the substitution of the negatively charged Asp265 residue of loop 2 with an Asn residue (Fig. 6a), as the ion·dipole interactions involving Asp265 found in FabB or FabF are substituted with weaker dipole·dipole interactions in actKS. Interestingly, in KSs from non-iterative type II polyene PKSs, such as Iga11[50,51] and ApeO[52], that require

substrate offloading in-between condensation steps[52], the same substitution is not observed, instead these enzymes possess the conserved Asp265 found in type II FAS KSs.

**Mutagenesis studies of FabF gating mutants.** To determine if the proposed gating function of loops 1 and 2 have any effect on KS activity, we identified a panel of FabF mutations that we hypothesized would attenuate FabF's ability to accept and/or process acyl-AcpP substrates. These mutations were designed to alter FabF's proposed gating function by modulating loop 1's flexibility (flex modulation), destabilizing (disfavoring) the gate-open conformation (destabilization), or occluding the pocket occupied by Phe400 in the gate-open conformation (pocket blockage). To test this hypothesis, we adapted a crosslinking assay

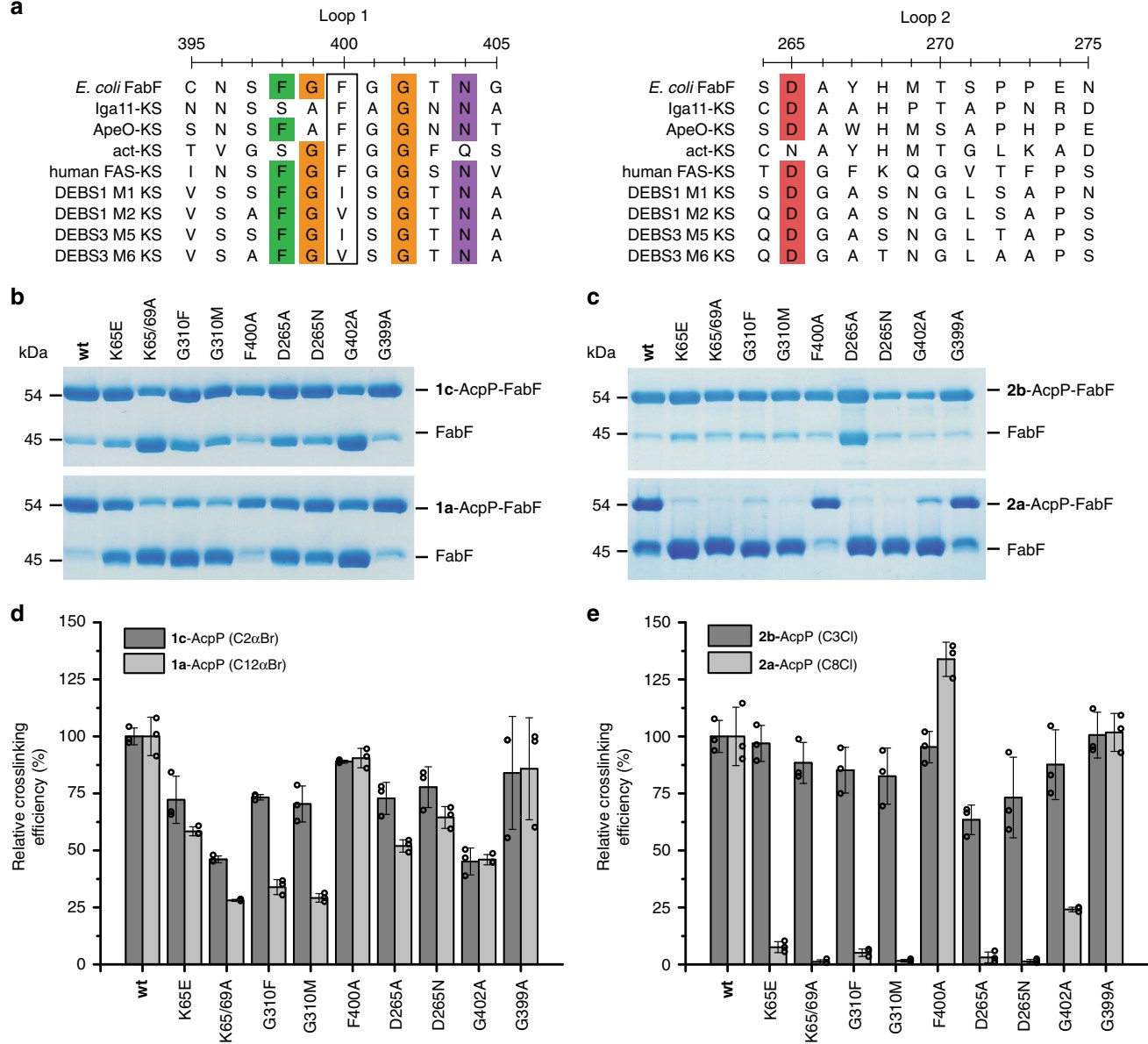

**Fig. 6 Loop 1 and 2 Sequence conservation and mutagenesis. a** Sequence alignment of loops 1 and 2 of FabF with representative KS sequences from non-iterative type II PKS (Iga11, ApeO), iterative type II PKS (act-KS), type I FAS (Human-FAS-KS), and type I PKS (DEBS KSs). The putative gating residue (Phe400 in FabF) for each synthase is outlined in a black box. **b** Single time point crosslinking gels of **1c**-AcpP (C2αBr-AcpP) and **1a**-AcpP (C12αBr-AcpP) with FabF gating mutants. **c** Single time point crosslinking gels of **2b**-AcpP (C3Cl-AcpP) and **2a**-AcpP (C8Cl-AcpP) with FabF gating mutants. **d** Densitometric analysis of single time-point crosslinking efficiency of **1c**-AcpP (C2αBr-AcpP) and **1a**-AcpP (C12αBr-AcpP) with FabF mutants. **e** Densitometric analysis of single time-point crosslinking efficiency of **2b**-AcpP (C3Cl-AcpP) and **2a**-AcpP (C8Cl-AcpP) with FabF mutants. All crosslinking experiments from **b** and **c** were performed as biologically independent experiments ($n = 3$) and all data represented in **d** and **e** are the average crosslinking efficiency of each mutant normalized to the average of wt FabF crosslinking efficiency. The error bars in **d** and **e** are represented as standard deviation (±SD) and the individual normalized measurements from each independent experiment are overlaid on top of the associated bar plot as open circles. Source data for all experiments are provided as a Source Data file.

that utilizes α-bromo and chloroacrylate mechanistic crosslinkers with different fatty acid chain lengths as substrate mimetics[39] (Fig. 2a). We hypothesized that our panel of FabF gating mutants would generally retain wild-type (wt) activity toward smaller C2 and C3 chain length crosslinkers, **1c** and **2a**, but would crosslink less efficiently with longer C8 and C12 chain length crosslinkers, **2b** and **1a**, which would require transitions between gate conformations to access the KS active site.

Before assaying these gating mutants, we first evaluated the crosslinking activity of a gate removal mutant, F400A. The results show that both F400A and wt FabF crosslink with generally equal

efficiency with short- and long-chain acyl-AcpP substrates (Fig. 6). Interestingly, the F400A mutant underwent crosslinking with the C8Cl (**2b**) with higher relative efficiency to wt (133%) both in our single time point assay and at a faster rate in our time-course crosslinking studies (Supplementary Figs. 6–11). These results suggest that mutating Phe400, a key gating residue[27], to alanine enhances the accessibility of probe to the KS active site cysteine[27]. In addition, we prepared two FabF interface mutants, K65E and K65A/K69A, to demonstrate that crosslinking can also report on PPIs. Both of these mutants showed reduced crosslinking towards all probes, but crosslinking

efficiency was most significantly reduced with long-chain cross-linkers (Fig. 6). These results suggest that well-coordinated PPIs are important for chain-flipping[53] substrates into the KS active site.

To prevent loop 1 transitions from the closed to open conformation, we mutated FabF's Gly310 to a phenylalanine or methionine residue to introduce a bulky amino acid at a position that would occlude the pocket occupied by Phe400 in the open conformation (pocket-blockage, Supplementary Fig. 12). Interestingly, G310F and G310M crosslink with 73 and 70% efficiency with C2αBr (1c) but only 33 and 30% efficiency with C12αBr (1a), respectively. Differential crosslinking activity between short and long substrates was more pronounced for the chloroacrylate crosslinkers as G310F and G310M crosslinked with 85 and 82% efficiency with C3Cl (2a) but only 5 and 1% efficiency with C8Cl (2b), respectively (Fig. 6). These results indicate that blocking the open conformation reduces FabF's ability to accept longer acyl-ACP substrates but not that of smaller substrates that more closely resemble malonyl-AcpP.

Similar patterns emerge when examining the crosslinking efficiencies of our D265A and D265N mutants of loop 2. These mutants are hypothesized to disfavor the open conformation by replacing the ion·dipole interactions involving the conserved Asp265 from loop 2 with weaker interactions. These mutations should therefore disrupt the conserved D265 coordinated hydrogen-bonding network between loops 1 and 2 that stabilizes the open conformation. D265A and D265N mutants crosslinked with 72 and 77% efficiency with C2αBr (1c) and 51 and 64% efficiency with C12αBr (1a), respectively. Furthermore, D265A and D265N crosslinked with 63 and 73% efficiency with C3Cl (2a) but only 2 and 1% efficiency with C8Cl (2b), respectively.

In contrast, loop flexibility-modulated mutants, G399A and G402A, did not demonstrate pronounced differences in cross-linking relative to wt FabF in our assay. The G399A mutant reacted with all crosslinkers with similar efficiency as wt FabF, while crosslinking efficiencies of the G402A mutant only showed differential crosslinking activity when comparing C3Cl (2a) (87%) and C8Cl (2b) (24%) (Fig. 6). G399A also crosslinked similarly to wt FabF in our time-course crosslinking assay (Fig. S5). While it was expected that the G399A mutant would show a difference in crosslinking efficiency, it is important to note that such a substitution is observed in nature, as the KSs from type II polyene synthases, Iga11[50,51] and ApeO[52], both have alanine at this position (Fig. 5a). In addition, the dihedral angles of G399A in both the closed and open conformation are not in the Ramachandran forbidden zone (Fig. 5b, c) and can therefore be reasonably accessed by the G399A mutant.

**MD simulations support a KS gating mechanism.** Using these AcpP–KS crystal structures as initial coordinates, we performed MD simulations of C10, C12, and C16-AcpP·KS complexes, where "·" denotes a noncovalent complex (Supplementary Figs. 1 and 13), to probe the dynamic properties of the gating processes in acyl-AcpP·KS complexes. Models of the ACP-bound acyl-AcpP·KSs were constructed by manually replacing the cross-linkers present in the crystal structures with the natural saturated substrates encountered by the KSs (Online Methods). In addition, simulations of apo-FabB and apo-FabF were performed using previously reported crystal structures (PDB: 2VB9[54] and 2GFW[43], respectively) as initial coordinates. Lastly, in an attempt to computationally visualize the transition between the open and closed conformations of loops 1 and 2, we performed additional MD simulations of apo-FabF and apo-FabB using KS structures prepared by removing the AcpP monomers from the AcpP–KS complexes reported herein. We distinguish these structures from

the crystallographic apo-FabB and apo-FabF structures by refer-ring to them as apo-FabB* and apo-FabF*. Because these apo-structures feature their loops in conformations that may not be stable in the absence of an associated acyl-ACP, we hypothesized that they may more readily undergo conformational transitions. Each of the ten systems described above was subjected to three independent 500 ns MD simulations. Replicates were initiated using different velocities and a total of 15 μs of MD data were collected. General analysis for all simulated systems is provided in Supplementary Figs. 14–27 and additional discussion regarding substrate interactions is provided in Supplementary Notes 6 and 7.

From this data, no transition between the closed and open loop conformations was sampled during the course of any simulation, suggesting that the coordinated movement of loops 1 and 2 between their open and closed states is a relatively slow process, likely occurring on timescales of tens or hundreds of μs[55]. Nonetheless, analysis of the per residue backbone and side chain root mean square fluctuations (RMSFs) of the loop residues, shown in Fig. 7d–i and Supplementary Figs. 28–30, suggests the following: Firstly, while the loops of apo-FabB and apo-FabF have similar RMSFs, the loops of apo-FabF* and the acyl-AcpP·FabF complexes show greater fluctuations than do those of apo-FabB* and the acyl-AcpP·FabB complexes, respectively. Of the structures shown in Fig. 7d–i, apo-FabF* is the most significant. RMSF analysis shows that loop 2 of apo-FabF* fluctuates more in this structure than any other simulated structures. Despite this dynamism, apo-FabF*'s loop 1 remains fairly rigid. It is possible that the transition of loop 2 from its open to closed conformation first requires the motion of loop 1 toward the active site. In fact, this movement of loop 1 would restore the oxyanion hole absent in the C16AcpP–FabF structure (Fig. 3). Therefore, the large RMSFs observed for apo-FabF*s loop 2 reveal a so-called frustrated loop unable to assume its preferred open conformation in the absence of acyl-AcpP (Fig. 7). These findings, along with published studies of PKS ACPs[16], suggest that ACPs may not only be responsible for substrate delivery but also may induce loop motions that promote catalysis through allostery.

Identification of this frustrated loop suggests that the motions of loops 1 and 2 are correlated. To evaluate this hypothesis, we determined the average correlations of the motions of residues of loops 1 and 2 sampled during the course of the MD simulations of apo-KS, apo-KS*, and acyl-AcpP·KS structures. Both FabB and FabF appear to form the same general network of interloop interactions to choreograph loop motion (Fig. 8). Nonetheless, the co-relatedness of the loops are distinct in the open (Fig. 8c, e) and closed (Fig. 8a–d) states, with a greater correlation found in the closed conformation. Furthermore, the average correlation of loop 1 and 2 residues is slightly greater in simulations of acyl-AcpP·FabF complexes than simulations of either apo-FabF or apo-FabF*. On the basis of this analysis, it is evident that loops 1 and 2 move in concert with one another and that the GFGG hinge motif is found to be most correlated to the highly conversed Asp265 residue of loop 2. Interestingly, these motifs are more highly correlated in FabF than in FabB, although higher positive average correlations between Asp265 and intra-loop 2 residues Pro271, Ser272, and Gly273 are observed in simulations of apo-FabB and its acyl-AcpP complexes.

The residues of loop 1 and 2 of FabB and FabF, regardless of enzyme state, that are most strongly correlated are residues of the GFGG motif of loop 1 and Asp265 of loop 2. In fact, the closed and open conformations of these loops can be clearly distinguished by differences in these intra- and interloop interactions. That we do not observe changes in these networks of interactions within any KS simulation indicates the stability of the open and closed states. According to simulation data, the

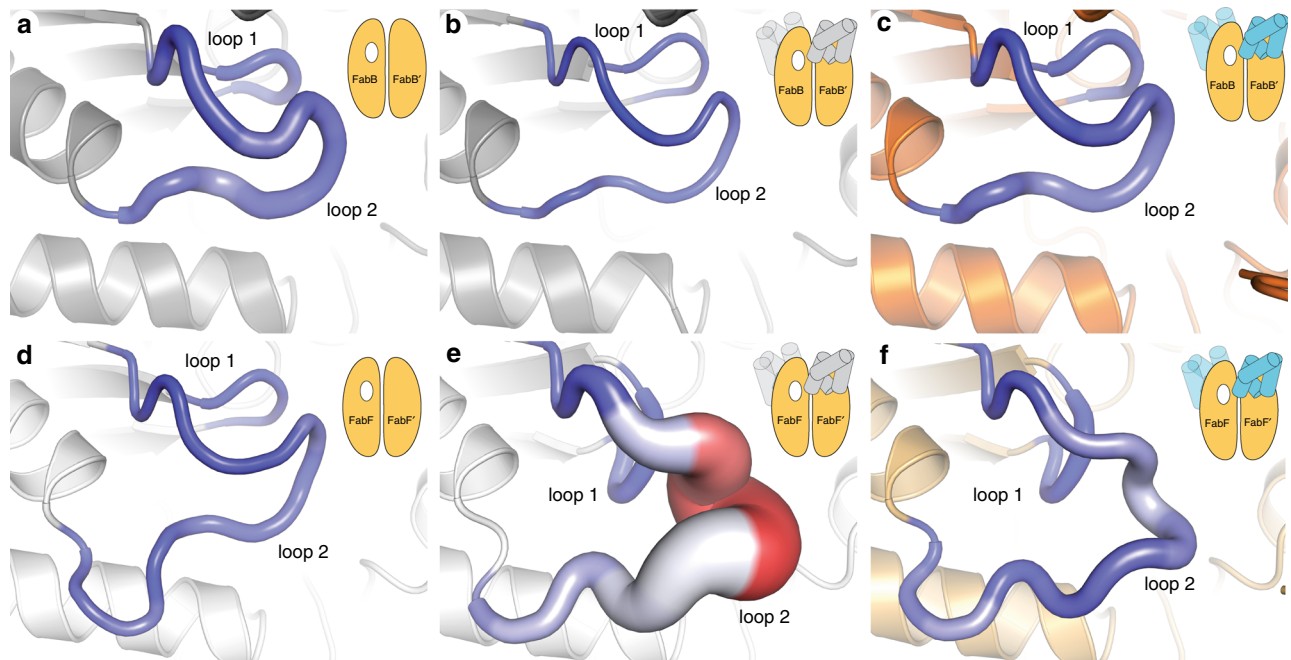

**Fig. 7 Loop dynamism sampled via computer simulations.** Per residue root mean square fluctuations (RMSFs) of **a**, *apo*-FabB₂; **b** *apo*-FabB₂* (*apo* structure derived from AcpP₂–FabB₂ structure); **c** AcpP₂·FabB₂; **d** *apo*-FabF₂; **e** *apo*-FabF₂* (*apo* structure derived from AcpP₂-FabF₂ structure); **f** AcpP₂·FabF₂. Larger (per residue) backbone RMSF values correspond to a thicker "sausage"; color range (blue to white to red) illustrates (per residue) side chain RMSFs with the blue-to-red color range indicating small-to-large RMSF values.

open conformation (apo-FabF* and acyl-AcpP·FabF) is characterized by hydrogen-bonding interactions involving the GFGG motif and Asp265 and Asn404. Namely, Gly399 forms a hydrogen bond with Asn404 while the negatively charged Asp265 residue simultaneously coordinates Asn404 and the backbone amides of all residues of the GFGG motif. In the open conformation of FabF, average distances of 5.09 Å, 3.91 Å, 3.66 Å, and 4.12 Å, respectively, are computationally observed between the center of mass of the carboxylate moiety of Asp265 and the backbone amide nitrogens of Gly399, Phe400, Gly401, and Gly402 (Fig. 9). Given the highly conserved nature of Asp265 and its substitution with a less potent hydrogen-bond acceptor (e.g., Asn265 in actKS) in iterative type II KSs, it is reasonable to postulate that the coordination of the motion of loop 1 and loop 2 is mediated by Asp265.

## Discussion

KSs catalyze multi-step reactions that require the association and dissociation of two distinct acyl-AcpPs and involve the intermediacy of an acyl-enzyme adduct. Enzymes that catalyze such complex chemical transformations often implement gating mechanisms to control solvent access, exert mechanistic selectivity, and/or control reaction order[37]. Zhang et al. previously showed that Phe400 of *Streptococcus pneumoniae* FabF (SpFabF) acts as a KS gating residue[27]. The F400A mutant of SpFabF has reduced catalytic activity relative to wt for its native reaction, but efficiently produces triacetic acid lactone, a common shunt product of condensing enzymes that results from the mispriming and extension of malonyl-ACP. These results suggest a compromised gating system that no longer controls reaction order. The proposed gating role of Phe400 is also supported with previously published structural data[27,43,56]. In structures of *apo*-FabB and *apo*-FabF, Phe400 initially blocks access to the active site cysteine but after transacylation rotates away to form the malonyl-ACP binding site for the pong half-reaction[27,43,56].

Results reported herein not only confirm and build upon the role of Phe400 as a KS gating residue but also provide additional context for its mechanism and regulation. With this information, we propose that elongating KSs use a double drawbridge-like gating mechanism[37] to direct substrate binding and control the timing and sequence of the transacylation and condensation half-reactions. From this study we propose the model shown in Fig. 10. Here, ACP binds to *apo*-KS, upon which ACP chain-flips[53] its cargo into the KS active site as KS loops 1 and 2 move in a coordinated manner to assume an open conformation, forming a transient, hydrophobic delivery/extraction pocket that temporarily accommodates the substrate (Fig. 10b). Loops 1 and 2 then return to their closed conformations, directing the substrate's acyl chain into the acyl binding pocket while simultaneously restoring the oxyanion hole that facilitates transacylation (Fig. 10c). Upon transacylation, *holo*-ACP dissociates and Phe400 rotates to form the malonyl-ACP binding pocket[22] (Fig. 10d).

The proposed gating mechanism coordinated by loops 1 and 2 is in agreement with our mutagenesis studies, MD simulations, and MSA analysis. FabF pocket-blockage mutants, G310M and G310F, and loop 2 destabilization mutants, D265A and D265N, show reduced crosslinking with long-chain acyl-AcpP substrates as compared with short-chain substrates. These results, taken together, indicate that blocking transitions to the gate-open conformation or disfavoring the gate-open conformation by disruption of the the interloop interaction network involving impairs KS activity with acyl-AcpPs. Interrogation of these trapped catalytic states using long time-scale MD simulations demonstrates that the open and closed conformations of FabF are stable and coordinated by a conserved network of hydrogen-bonding interactions. Furthermore, this network of interactions serves to correlate the motions of loops 1 and 2 and is generally conserved in elongating KSs in type I PKS, type I FAS, and type II PKS.

Given our gating model (Fig. 10), we now propose a general mechanism used by these synthases to regulate substrate processing

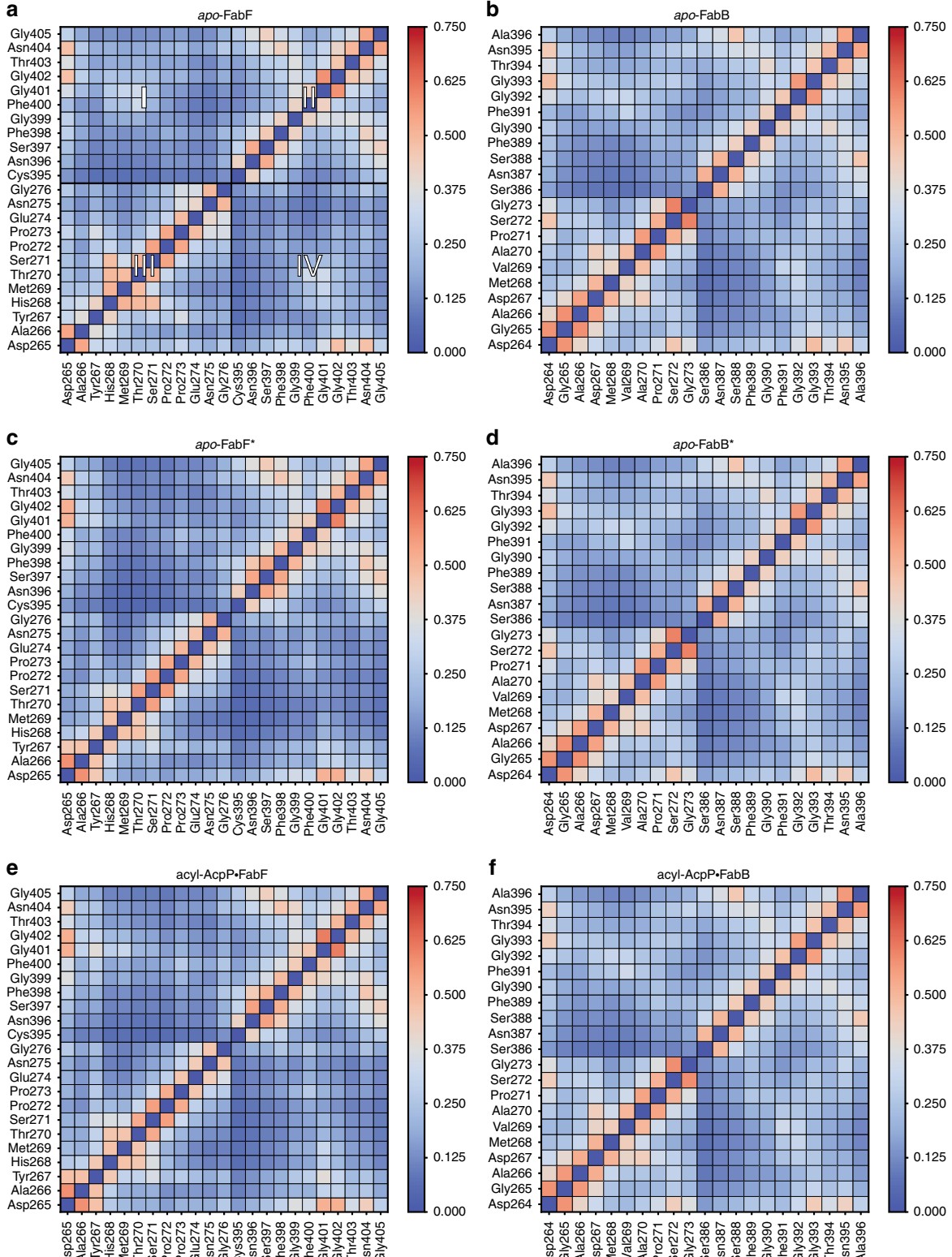

**Fig. 8 Average cross correlations of the motions of key loop 1 and 2 residues sampled computationally.** Cross correlations were determined by computing a motion vector for each residue of loop 1 and loop 2 from its prior position to its current over the course of each simulation. In this manner, motion vectors for each pair of residues of loop 1 and 2 are calculated for each frame of simulation data. Heat maps generated from the cross-correlation analysis of the motions of loop 1 and 2 residues of **a** *apo*-FabF, **b** *apo*-FabB, **c** *apo*-FabF*, **d** *apo*-FabB*, **e** acyl-AcpP·FabF, and **f** acyl-AcpP·FabB, respectively, sampled using molecular dynamics (MD) simulations. **a** The heat map has been divided into four quadrants; quadrants II and III illustrates cross correlations of the motions of residues within loops 1 and 2, respectively. Inter-loop cross correlations are shown in quadrants I and IV. Note that each heatmap is symmetric about the diagonal that bisects quadrants II and III.

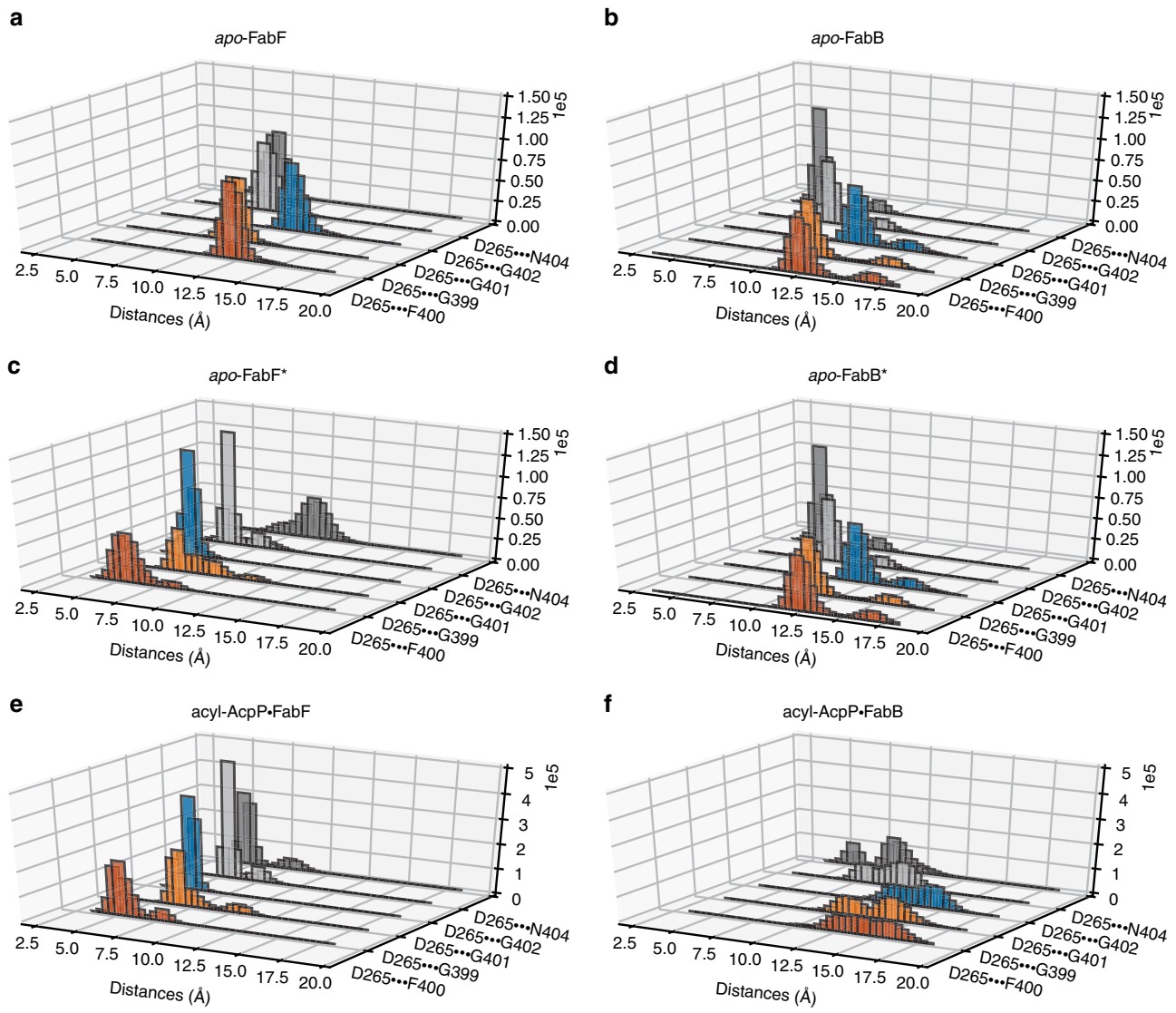

**Fig. 9 Analysis of the fluctuations of inter-loop hydrogen bonding networks.** Histograms showing the distances between the hydrogen-bond donating Asp265 and highly conversed (hydrogen-bonding accepting) resudes, Gly399, Phe400, Gly401, Gly402, and Asn404 sampled computationally. Histograms of these distances sampled in molecular dynamics (MD) simulations of **a** apo-FabF, **b** apo-FabB, **c** apo-FabF*, **d** apo-FabB*, **e** acyl-AcpP·FabF, and **f** acyl-AcpP·FabB, respectively.

and catalysis. Transition to the open-conformation disrupts the oxyanion hole, which suggests that (trans)acylation of the active site cysteine occurs after the gate adopts the closed conformation. Consequently, the gate could act as a sensor that only reorganizes the oxyanion hole when the correct substrate is bound. Therefore, the gate and F400 may prohibit FAS KSs from reacting with fatty acyl intermediates that are not appropriate condensation substrates (i.e., β-hydroxy-acyl-ACPs).

A careful consideration of loops 1 and 2 and their potential specific functions reveals some interesting insights as well. Ramachandran analyses of the GFGG motif's fluctuations over the course of MD simulations illustrate the rigidity of loop 1 in both the closed and open conformations. The φ and ψ angles of the GFGG residues sampled computationally are narrowly distributed with hinge residues, Gly399 and Gly402, possessing distinct backbone dihedral angles in the open and closed conformations (Fig. 5b, c). This is in contrast to Ramachandran analysis of the hinge residues in loop 2, which shows that although the φ and ψ angles of Pro272 and Pro273 are distinct in the open and closed states in our crosslinked structures, they

sample similar φ, ψ angle distributions during simulations (Fig. 5e, f and Supplementary Figs. 31 and 32). These results, along with the increased sequence variation in different condensing enzyme families, suggests loop 2 may modulate loop 1 gating events, potentially conferring different activities or substrate specificities to distinct KSs. This is an appealing hypothesis, as loop 2 sits on top of loop 1, effectively capping it in when in the closed conformation. Furthermore, the dynamism of loop 2 in our apo-FabF* simulations indicates that acyl-AcpP substrates stabilize the gate-open conformation. In fact, AcpP binding or the chain flipping of acyl cargo into the KS active site may serve to trigger gating events by stabilizing loop 2's open position. Movement of loop 2 would thereby allow loop 1 to assume the gate-open conformation, raising the drawbridge and allowing substrate access to the active site. While more work is needed to test this hypothesis, a recent study by Robbins et al. demonstrated that mutation of S315A from loop 2 of DEBS KS1 increased the formation of propionate, which results from the decarboxylation of the methylmalonyl-ACP substrate[57]. These findings suggest an important role for loop 2 residues in DEBS and that the altered

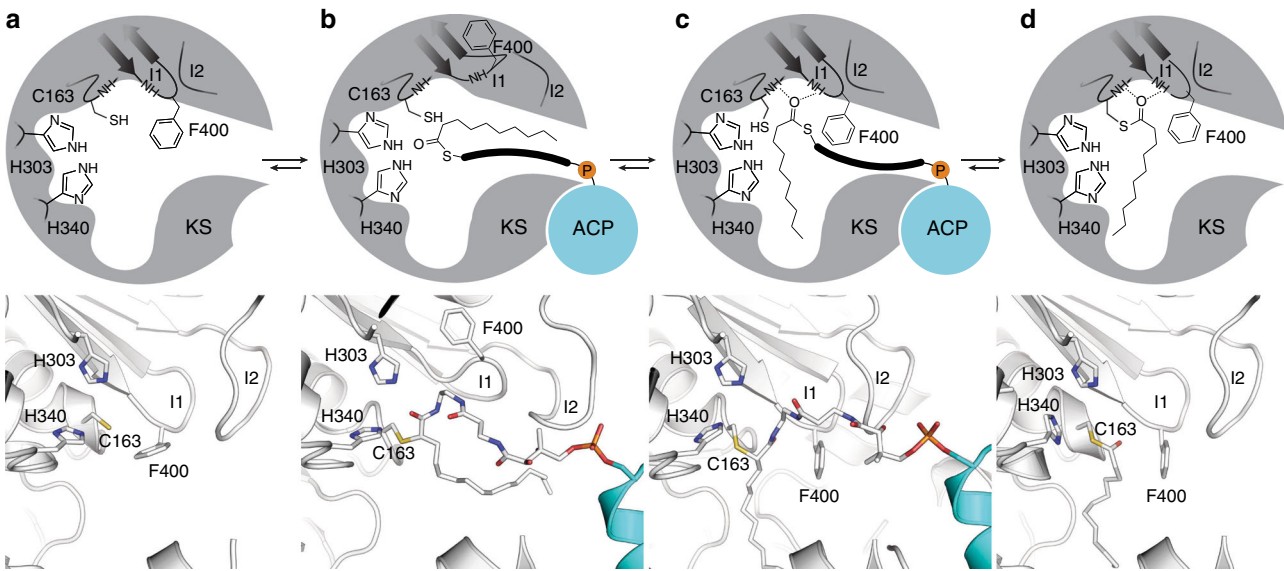

**Fig. 10 Proposed gating mechanism of elongating ketosynthases.** Figure demonstrates the active site conformational changes that facilitate acyl-chain transfer in elongating KSs as demonstrated by trapped crystallographic states from this work and previous studies. **a** The *apo*-form of the KS active site illustrated by a 2D schematic of the active site architecture (top) and the crystal structure of apo-FabF (PDB: 2GFW) (bottom). **b** Active site reorganization to the gate-open conformation upon association of acyl-AcpP with KSs as seen in the C16AcpP–FabF structure reported herein with a 2D schematic of the active site (top) and 3D rendering of the active site (bottom). **c** Active site of C12AcpP–FabB, illustrating the gate-closed transacylation competent form of the active sites of elongating KSs shown as 2D schematic (top) and 3D rendering (bottom). Residues are represented in FabF numbering. **d** The acyl-enzyme adduct form of FabF (PDB: 2GFY) shown as a 2D schematic (top) and 3D rendering (bottom). Active site catalytic residues Cys163, His303, His340, and Phe400 are provided in FabF residue numbering, represented as sticks, and colored according to element (C-white, O-red, N-blue, S-yellow). Hydrogen-bonding interactions are represented as dotted lines.

S315A mutant activity may be indicative of an impaired gating system.

The importance of KS-catalyzed carbon–carbon bond formation in biosynthetic pathways cannot be overstated. These enzymes perform complex reactions and employ poorly understood biochemical regulatory mechanisms to ensure substrate specificity and product fidelity. Here, the combination of chemical, structural, and computational biology together support a gating mechanism used by elongating KSs to recognize and process acyl-ACP substrates. These results provide broadly applicable insights into KS activity, function, and substrate selectivity as well as further our understanding of the *E. coli* AcpP interactome.

## Methods

**Ketosynthase production, purification, and tag cleavage.** The N-terminal His$_8$-tag FabF and FabB recombinant proteins were expressed in *E. coli* BL21 and cultured in Terrific Broth (MilliporeSigma). Cells were grown in the presence of 50 mg·L$^{-1}$ kanamycin, induced with 0.5 mM Isopropyl β-d-1-thiogalactopyranoside (IPTG) at OD$_{600}$ = 0.8, and incubated at 37 °C for 4 h. The cells were spun down by centrifugation at 500 × *g* for 30 min and the collected pellets were resuspended in lysis buffer (50 mM Tris, 150 mM NaCl, 10% glycerol, pH 8.0) along with 0.5 mg·mL$^{-1}$ lysozyme (Worthington Biochemical Corp). The pelleted cells were lysed by sonication (4 s pulses for 5 min), followed by another centrifugation at 17,400 × *g* for 30 min to clear the lysate. The proteins were purified using Ni-NTA resin (Thermo) in the column and were washed sequentially with wash buffer (50 mM Tris, 150 mM NaCl, 10% glycerol, pH 8.0) followed by elution with 250 mM buffered imidazole. The His$_8$-tags of purified proteins were cleaved with bovine thrombin (2 U per 1 mg protein) for 16 h at 6 °C while dialyzing against the dialysis buffer (50 mM Tris pH 8.0, 150 mM NaCl, 10% glycerol, 0.5 mM TCEP). Resulting solutions were re-purified using Ni-NTA resin (Thermo Fisher Scientific) to trap the un-cleaved proteins. The FabF and FabB were further purified by FPLC using the HiLoad Superdex 200 (GE Biosciences) size exclusion column. The eluted proteins were collected and concentrated to 2–4 mg·mL$^{-1}$ using Amicon Ultra Centrifuge Filters (MilliporeSigma) with 10 kDa molecular weight cut off.

**Native AcpP recombinant protein purification.** Native AcpP recombinant protein was expressed in *E. coli* BL21 (DE3) and grown in Terrific Broth (Source).

Cells were grown at 37 °C in the presence of 100 mg·L$^{-1}$ ampicillin, induced with 0.5 mM IPTG at OD$_{600}$ = 0.8, and incubated at 18 °C for 16 h. The cells were spun down by centrifugation at 500 × *g* for 30 min, and the collected pellets were resuspended in lysis buffer (50 mM Tris, 5% glycerol, pH 7.4) along with 0.1 mg·mL$^{-1}$ lysozyme (Worthington Biochemical Corp). The pelleted cells were lysed by sonication (4 s pulses for 5 min). Due to the high stability of AcpP, irrelevant proteins were precipitated by dripping in same volume of isopropanol into the lysate at the speed of 0.1 mL·s$^{-1}$[51]. The resulting mixture was spun down by centrifugation at 11,200 × *g* for 1 h and the supernatant was injected directly into FPLC. AcpP was purified by HiTrap Q HP anion exchange chromatography column in 50 mM Tris buffer with a gradient of NaCl from 0 M to 1 M and the native AcpP eluted around 0.3 M NaCl. The eluted protein was collected and concentrated using Amicon Ultra Centrifuge Filters (Millipore) with 3 kDa molecular weight cut off.

**Holo-AcpP apofication and apo-AcpP purification.** The purified protein was resulted in a mixture of *holo*-AcpP and *apo*-AcpP. The phosphopantetheine moiety on *holo*-AcpP was removed by acyl carrier protein hydrolase (AcpH) in lysis buffer for 16 h[58]. Final reaction concentrations: 5 mg·mL$^{-1}$ AcpP mixture, 0.01 mg·mL$^{-1}$ AcpH, 50 mM Tris (pH 7.4), 10% glycerol, 10 mM MgCl$_2$, 5 mM MnCl$_2$, and 0.25% DTT. The resulting pure *apo*-AcpP, was analyzed by conformationally sensitive urea PAGE (2.5 M urea), and further purified by FPLC using HiLoad 16/600 Superdex 75 PG (GE Biosciences) size exclusion column to remove AcpH. The eluted protein was collected and concentrated using Amicon Ultra Centrifuge Filters (Millipore Sigma) with 3 kDa molecular weight cut off.

**Apo-AcpP loading and crypto-AcpP purification.** All crosslinkers were loaded onto *apo*-AcpP using a one-pot chemoenzymatic method[38]. This method utilizes three CoA biosynthetic enzymes (CoaA, CoaD, CoaE) to form CoA analogues and a phosphopantetheinyl transferase (PPTase, Sfp) to load them onto *apo*-AcpP, resulting in *crypto*-AcpP. Final reaction concentrations: 1 mg·mL$^{-1}$ *apo*-AcpP, 0.04 mg·mL$^{-1}$ Sfp, 0.01 mg·mL$^{-1}$ CoaA, 0.01 mg·mL$^{-1}$ CoaD, 0.01 mg·mL$^{-1}$ CoaE, 50 mM potassium phosphate pH 7.2, 12.5 mM MgCl$_2$, 1 mM DTT, 0.2 mM pantetheineamide probe (**1a,1c** or **2a,2b**) (Fig. 2), and 8 mM ATP. The stock solutions of crosslinkers **1a-1c**, **2a**, and **2c** were prepared by dissolving them in DMSO to a final concentration of 50 mM. Reactions were incubated at 37 °C for 16 h and then purified by HiLoad Superdex 75 (GE Biosciences) size exclusion column. The eluted protein was collected and concentrated using Amicon Ultra Centrifuge Filters (Millipore Sigma) with 3 kDa molecular weight cut off up to a concentration at 1–3 mg·mL$^{-1}$.

**Crosslinking reaction and crosslinked complex purification**. The crosslinking reactions were carried out by mixing *crypto*-AcpP with its partner protein (FabF or FabB) in 5:1 ratio at 37 °C for 16 h. Reactions were examined by 12% SDS PAGE and purified by HiLoad 16/600 Superdex 200 PG (GE Biosciences) size exclusion column using minimal buffer (20 mM Tris, 50 mM NaCl, pH 7.4). The resulting protein complex was greater than 95% in purity, determined by SDS PAGE, and was concentrated to 8–10 mg·mL⁻¹ using Amicon Ultra Centrifuge Filters (MilliporeSigma) with 30 kDa molecular weight cut off. The concentrated crosslinked complex was immediately used for protein crystallization or flash-frozen and stored in −80 °C freezer for later use.

**Construction of ketosynthase mutants**. Ketosynthase mutants were generated using the site directed mutagenesis method developed by Liu and Naismith[59]. All constructs were verified via Sanger sequencing (Eton Bioscience Inc.). Primers used for mutagenesis can be found in Supplementary Fig. 33.

**Crosslinking assay**. Each reaction was set up to contain 40 μM of crypto-AcpP, 20 μM of FabF, and buffer (25 mM Tris, 150 mM NaCl, 10% glycerol, 0.5 mM TCEP, pH 7.4). To stop the reaction at the designated time point, 2.5 μL of the reaction solution was mixed with 10 μL of 3X SDS dye. Samples were analyzed by 12% SDS-PAGE and the intensity of protein bands was measured by ImageJ[60]. The crosslinking efficiency was calculated by $[(I_{crosslink})/(I_{crosslink} + I_{FabF})] \times 100\%$ where "$I$" stands for band intensity. All reactions were performed in biological triplicate and the crosslinking efficiency of mutants was normalized to the average wild type crosslinking efficiency to obtain a normalized relative crosslinking efficiency. The plotted data represent the average normalized crosslinking efficiency of each mutant and the error bars are represented as standard deviation (±SD).

**General synthetic methods**. General synthetic methods are provided in Supplementary Note 8 and NMR characterization of compounds and intermediates is provided in Supplementary Figs. 40–43.

**Crystallization, structure determination, and refinement**. The crystals of the crosslinked complexes were grown by vapor diffusion at 6 °C. In detail, 1 μL of crosslinked complex (8–10 mg·mL⁻¹) was mixed with 1 μL of corresponding crystallographic condition and the mixture was placed inverted over 500 μL of the well solution (hanging-drop method). The AcpP–FabF complexes crystallized in 26–30% PEG 8 K, 0.1 M sodium cacodylate pH 6.5, and 0.3 M NaOAc, yielding numerous orange rods amongst a heavy precipitate. Seeding trials were performed to increase size and quality of the samples. The C12AcpP–FabB complexes produced crystals in two conditions: 20% PEG 8 K, 0.2 M Mg(OAc)₂, 0.1 M sodium cacodylate pH 6.5 as well as 30% PEG 8 K, 0.2 M (NH₄)₂SO₄, 0.1 M sodium cacodylate, pH 6.5. Both conditions yielded square plates and required 1–2 weeks for complete growth of crystals. The C16AcpP–FabB complexes produced crystals in 18–24% PEG 8 K, 0.1 M sodium cacodylate, and 0.3 M NaOAc, pH 6.5.

All data were collected at the Advanced Light Source synchrotron at Berkeley. Data were indexed using iMosflm[61] then processed and scaled using aimless from the CCP4 software suite[62,63]. Scaled reflection output data were used for molecular replacement and model building in PHENIX[64]. For AcpP–FabF complexes, phases were solved using molecular replacement by first locating the larger FabF (PDB: 2GFW[43]) monomer followed by a second search function to place the smaller AcpP (PDB: 2FAC[65]) molecule in the residual density. For C12AcpP–FabB and C16AcpP–FabB complexes, initial phases were solved using molecular replacement by first performing a search function using the FabB monomer (PDB: 1G5X[30]) and then manually placing the AcpP molecules in the residual electron density. The parameter file for the covalently bonded 4′-phosphopantetheine was generated using Jligand (CCP4)[63]. Manually programmed parameter restraints were used to create the associated covalent bonds between 4′-phosphopantetheine to Ser36 and Cys163 during refinement. These parameters are provided in the parm-file folder located in the Source Data file.

**Preparation of *apo*-KS structures for MD simulations**. Simulations of *apo*-FabB and *apo*-FabF were performed using two sets of initial coordinates: one derived from previously reported 1.55 Å *apo*-FabB (PDB: 2VB9[54]) and 2.40 Å *apo*-FabF (PDB: 2GFW[43]) and a second set generated by modifying the crosslinked structures of *E. coli* AcpP–FabB and AcpP–FabF reported herein by deleting coordinates corresponding to atoms from AcpP.

**Preparation of acyl-AcpP·KS structures for MD simulations**. The crosslinked structures of *E. coli* AcpP–FabB and AcpP–FabF reported herein were used to prepare the initial coordinates used to perform MD. The asymmetric unit of the AcpP–FabF structure is—unlike that of crosslinked AcpP–FabB—the 1:1 AcpP·FabF complex. In order to prepare coordinates for the (functional) biological unit, AcpP₂·FabF₂, the asymmetric unit of the AcpP₁–FabF₁ structure was rotated about the twofold symmetry axis of the AcpP–FabF biological unit. The 2:2 complexes of acyl-AcpPs shown in Scheme S1 and either FabB or FabF were constructed by manually modifying the chemical crosslinker present in the experimental structures. Modifications of this moiety were performed using

Avogadro 1.1.1[66] [http://avogadro.cc/], Gaussview 5.0.9, and Pymol v.1.8.6, and Pymol v.2.2.3. Schrodinger was used to add poorly resolved C- and N-terminal residues omitted in the dimeric AcpP·FabB and AcpP·FabF complexes. The protonation state of titratable residues in the *apo*-KSs and acyl-AcpP·KS didomains were assigned using the H++ webserver, with the exception of the active site Cys163 of FabB and FabF; this residue was simulated in as a thiolate[67–70]. Histidine protonation states were inspected manually.

**Parameter generation for acyl-AcpP**. For each simulated AcpP·KS complex, the acyl substrate, phosphopantetheine cofactor, and the conserved serine residue of AcpP to which the cofactor is affixed were treated as a single nonstandard residue. AMBER (ff14SB[71]) and GAFF[72] type force field parameters were assigned to the atoms of these nonstandard residues using ANTECHAMBER[73]. Partial charges for all atoms in all nonstandard residues were determined using the RESP methodology[74]. All quantum calculations were performed using Gaussian 09 (https://gaussian.com/glossary/g09/).

**Preparation of simulation cells**. Proteins were solvated in a water box with TIP3P[75] water molecules. Using TLEAP[76], the simulation cell was constructed such that its edges were placed 10 Å away from the closest proteinogenic atom. TLEAP was used to neutralize and salt the water box to mimic physiological conditions (0.15 M). Counterions (Na⁺ and Cl⁻) were also added to the simulation cell randomly.

**Simulation details**. Amber16 and Amber18[76] were used to perform all MD simulations. All simulations were performed using the ff14SB[71] and GAFF[72] force fields. A 2 fs time-step was utilized via the SHAKE algorithm, which constrains all nonpolar bonds involving hydrogen atoms[77]. Long-range electrostatic interactions were treated using the Particle Mesh Ewald (PME) method with a 10 Å cutoff for all non-bonded interactions[78]. Both solvated protein complexes were energy minimized in a two-step fashion. In a first step, solvent molecules and counterions were allowed to relax, while all protein atoms were restrained using a harmonic potential ($k = 500$ kcal mol⁻¹ Å⁻²). This geometry optimization was followed by an unrestrained energy minimization of the entire system. The thermal energy available at a physiological temperature of 310 K was slowly added to each system over the course of a 3.5 ns NVT ensemble simulation. The solvated complexes were then subjected to unbiased isobaric–isothermal (NPT) simulations for 25 ns in order to equilibrate the heated structures. Three independent 500 ns production MD (NPT ensemble) of each system were performed with different initial velocities. For both NVT and NPT simulations, the Langevin thermostat ($\lambda = 5.0$ ps⁻¹) was used to maintain temperature control[79]. Pressure regulation in NPT simulations (target pressure of 1 atm) was achieved by isotropic position scaling of the simulation cell volume using a Berendsen barostat.

**Analysis and visualization of simulation data**. Analysis was performed using CPPTRAJ Version V4.14.0 (AmberTools V19.00), PYTRAJ v2.02.dev0, a Python front-end for the CPPTRAJ analysis code, and MDTRAJ v1.9.3[80,81]. Trajectories were visualized using VMD 1.9.4[82] and Pymol v1.8.6 and Pymol v2.2.3. Coordinate data were written to disk every 10 ps.

**Reporting summary**. Further information on research design is available in the Nature Research Reporting Summary linked to this article.

## Data availability

All crystallographic models and data were submitted to the Worldwide Protein Data Bank (wwPDB, www.wwpdb.org) under the accession codes 6OKC (C12AcpP–FabB), 6OKF (C16AcpP–FabB), 6OLT (C12AcpP–FabF), 6OKG (C16AcpP–FabF). All parameter files for generating and running MD simulations are provided in the parm-files folder in the online Source Data file. All original source data used for crosslinking analysis reported in Fig. 6 and Supplementary Figs. 5, 6, and 8 are provided as a Source Data file. All other data are available from the corresponding authors on reasonable request.

## Code availability

All MD simulation (AMBER 16 and AMBER 18) and analysis (PYTRAJ, CPPTRAJ) code used herein is published and publicly available.

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

## Acknowledgements

This research was supported by National Science Foundation Grant EEC-0813570 (to J.P.N.) and NIH GM095970 (to M.D.B.). NIH grant T32 GM008326 for support of J.T.M. NIH Grant T32 GM112584 for support of T.G.B. T.D.D. is a San Diego IRACDA Postdoctoral Fellow supported by NIH K12 GM068524. Portions of the work were also funded by the Arthur and Julie Woodrow Chair at the Salk Institute (to J.P.N.) and the Howard Hughes Medical Institute (to J.P.N.). The authors thank Dr Gordon Louie for assistance in x-ray data collection, processing, and refinement and Marianne Bowman for assistance with protein crystallization. The authors also thank Dr Laetitia Misson for assistance with data analysis. We further acknowledge the following organizations for support of the computer simulation work reported herein: NIH GM31749 (to J.A.M.) and the San Diego Supercomputing Center (to J.A.M.).

## Author contributions

J.J.L., J.A.M, M.D.B., and J.P.N. designed and supervised the project. T.D.D. synthesized probe molecules. J.T.M, W.E.K., T.G.B., and A.C. purified protein, optimized cross-linking, carried out crystallization, and obtained crystals. J.T.M. collected x-ray diffraction data and refined and analyzed crystallographic models. J.T.M designed and produced gating mutants. W.E.K. and A.C. performed crosslinking assays. A.P. performed MD simulations. A.P., J.T.M., and J.A.M analyzed simulation data. J.T.M., A.P., J.J.L., J.A.M., M.D.B., and J.P.N. wrote the manuscript. All authors edited the manuscript. W.E.K., T.D.D., A.C., and T.G.B. contributed equally to this work.

## Competing interests

The authors declare no competing interests.
