## [Peer Review File · Nature Communications]

Reviewers' comments:

Reviewer #1 (Remarks to the Author):

Mindrebo et al. present a series of X-ray crystal structure of the enzyme KS in complex with the carrier domain ACP from the *E. coli* fatty acid synthase (FAS) type II system. This paper is a new research highlight from the Burkart lab to characterize substrate shuttling in fatty acid synthesis. In this manuscript, Mindrebo et al. have stalled ACP at the KS by using the specific crosslinking systems development, which has been developed during the last years. Here, ACP is modified by a warhead, transferred onto the ACP by action of the natural phosphopantetheine transferase reaction, which then covalently stalls the ACP at the enzyme. In this work, they use bromo-amido-acyl compounds, variable in length, so that chain length dependent binding can be structurally studied. Complexes of the ACP AcpP with the KS enzymes FabF and FabB are studied with acyl chains C12 and C16. FabF and FabB are the KS enzymes that are responsible for the elongation of the acyl-chain up to C16. The structural studies are supported by MD simulations in order to model domain-domain dynamics and acyl chain binding properties on the basis of the experimental data.

Data shared in this manuscript are highly exciting and surely deserve being shared with the broad readership of Nature Communications. In spite of the scientific impact, the difficulty in this manuscript in the current form lies in the very complicated presentation of data. Two main reasons make this paper difficult to read and understand. (i) Figures and Figure legends: Almost every figure panel shows structural details in a new view. It needs the careful reading of the legends to be able to understand which of the many structures is shown or which structures are superimposed. The sequence of figures does moreover not reflect the sequence of references in the text; e.g., see example to Figure 3 and 4 below. (ii) The timing of introducing information is suboptimal. There is partly too much data introduced simultaneously. The authors should check whether they can build up the surely exciting arguments in a more careful and logical manner; again highlighted below on the basis of one section. A revision of the manuscript with a particular focus on these two points would ensure a better understanding.

Difficulties in the readability of the manuscript are outlined in the following on the example of chapter: "Conformational heterogeneity in FabF active site loops":

Chapter "Conformational heterogeneity in FabF active site loops" starts introducing the C12AcpP-FabF structure, which lacks density of the acyl chain and suffers from poor density of loop 1 owing to conformational heterogeneity. Then it says that "we turned to the 2.30 Å C16AcpP-FabF dataset". Here the authors set a generic reference to Figure 3,4. There are 16 figure panels in Figure 3 and 4 – are all panels important here? Why not directly starting the chapter with C16AcpP-FabF data? The C12AcpP-FabF structure does not seem to be relevant here and may be introduced later or omitted at all. Even for describing phenylalanine (F400) movement presented in the following chapter "FabF active site loops adopt a "gate-open" conformation to accommodate acyl cargo", C12AcpP-FabF data is not required. The chapter "Conformational heterogeneity in FabF active site loops" finally closes

with MD modeling. The C16AcpP-FabF dataset shows carbonyl group of the fatty acyl chain resides within hydrogen-bonding distance of the two catalytic histidines. MD simulation based on this data obviously increase this distance suggesting that the carbonyl group is not forming hydrogen bonding with either active site histidine so that this state is a pre-transacylation state. At this point, authors refer to Fig. 3a-c. Why referring to X-ray data here? And which C16-chain is shown in Fig. 3c - C16a or C16b? Specific reference to panels 3d and 3e (except the generic "Figure 3,4 reference" mentioned above) appear much later in the text, after Fig.4 panels are introduced. The imprecise figure references as well as the structural information presenting in varying views make it very difficult to understand this section. Also the other parts of the manuscript suffer from poor readability.

Additional major points:

- Can the authors comment of the specific value of the structures C12AcpP-FabF and C16AcpP-FabB?
- The consensus on the function of F400/F392 in FabF/B (and other KS) is the sensing of the β -carbon modification so that just acyl chains with methylene are accepted for elongation. Data from Zhang et al are in line with such a function (ref 17). Do which extent does the gating mechanism go beyond this concept? Does the gating mechanism allow detecting the correct substrate prior to its elongating into the binding pocket? Or in other words, when does the gating mechanism distinguish between a productive and non-productive interaction (at the "gate open" position)? This is an important point, as the authors also mention in the introduction: "... Therefore, AcpP and its partner enzymes must minimize the number of non-productive PPIs". But this points remains vague.
- How well are the key arguments of this paper supported by data: (i) Why does structure C12AcpP-FabF differ from C12AcpP-FabB? Why does C16AcpP-FabB differ from C16AcpP-FabF? Maybe FabF and FabB are more different that anticipated here? (ii) The gating mechanism is built on mainly C12AcpP-FabB. One may also argue that C12AcpP-FabB is simply an artifact/local minimum structure received under the crystallization conditions? Is there more evidence for the gating mechanism? MD data may be supportive here. A better readability of the manuscript may easily clarify these points.

Other points:

- There are references to recent literature missing.
- The references to Figure panels are inaccurately placed (see also above).
- Why is the loop with residues range 399-402 numbered with loop 1 and the (N-terminal) loop with residues 263-275 loop 2. This is counter-intuitive.
- Residue Phe392 is incorrectly termed Phe400 in Chapter "Acyl-AcpP-FabB locks into a catalytically competent, "gate-closed". Please check numbering throughout the manuscript.

-Residues in Figure 6 should not be labeled in FabF residue numbering. It can be misleading when blurring the origin of data?

-Improve figure style: Reduce view. Reduces panel numbers; e.g., by connect panels (e.g. cartoon representations in Figure 5 can be shown as inset). Indicate "dataset" directly in the figure panel. Improve color code (show color code legend). ...

-X-ray structures should be made available during the review process (as review-only material).

Reviewer #2 (Remarks to the Author):

The authors have done a great thing for the community, in providing structural data as to how ACPs dock their cognate KSs. The development of crosslinkers, the purifications, and the structural elucidations represent a lot of hard work. However, my opinion is that the manuscript is not ready for publication. The authors may be reading into their structures too much. While the double-drawbridge gate is an interesting idea, only crystals of the ACP-FabF complexes show the movement of these loops. Despite the authors attempts, the proposed gating mechanism lacks molecular detail. Does FabF use this to select against partially processed acyl chains? How would KS/CLF's use these same motifs? An active site conformational change does not seem necessary in the function of these enzymes. The authors conduct molecular dynamics simulations, paying close attention to residues in these loops, but its not clear that they have learned much from these studies, which make up a considerable portion of the text. My large preference would be to replace this with functional assays of point mutants. For instance, how important is the threonine residue on loop 2? Is there an equivalent residue in FabB's loop 2? Site-directed mutagenesis studies to back up the claimed gating mechanisms are necessary (other ACP/KS interfacial residues could also be tested).

Reviewer #3 (Remarks to the Author):

This combined crystallography and computational study seeks to shed light on the mechanism of ketoacyl synthase by studying the interaction of E. coli FabF and FabB with substrate-bearing acyl carrier protein. The complexes are covalently trapped using bromo analogues and X-ray structures solved for C12 and C16 substrates with FabF or FabB. In the FabF-C16AcpP structure, an unusual orientation of two FabF loops are observed, in addition to an unusual substrate orientation, which the authors claim may represent a loop-based gating mechanism. Using sequence analysis, they

point to the conserved nature of the shorter loop and the variable nature of the second loop, suggesting a general mechanism for binding/catalysis but also tuned for specificity depending on the particular Fab protein involved. MD simulations are also employed (see later).

There are a couple of appealing and novel aspects to this work: it furnishes the first crystal structures of a KASII-type ketoacyl synthase (FabF) in complex with acyl-ACP and provides new structural insights into orientation of potential intermediates in fatty acid synthesis. It also proposes a clear structural hypothesis for a loop-based gating mechanism, supported by sequence analysis, and suitable for further testing in future studies. A more detailed understanding of the complex process of fatty acid synthesis is of broad interest across the fields of biology, biotechnology and medicine. This work also provides further evidence of protein conformational change as a key in dictating molecular recognition and reactivity.

In terms of the methods employed, based on my expertise I focus on the molecular dynamics simulations. However I first note that the X-ray structures are of adequate resolution, given careful interpretation of the density, to support the main conclusions of the work stated above. Indeed, comparison of apo-FabF and the C16AcpP-FabF crystal structures convincingly suggest that loop movement is required to accommodate the substrate (Figure 4a and p8). For the MD, the simulations are appropriately executed, with suitable force fields, equilibration and sampling using triplicate 500 ns simulations of each system. The methods and results for the most part are clearly presented (see below).

Therefore I recommend this novel contribution for publication in Nature Communications (subject to considering the comments below), as providing a valuable furthering of knowledge in the mechanism of fatty acid synthesis.

The purpose of the MD simulations within this work should be more clearly stated in the text. The MD simulations are applied primarily to extrapolate from the covalently trapped X-ray structure to the noncovalently bound substrate-Fab complex. It is found that only the 16:0-AcpP-FabB complex shows thiolate at the right distance for nucleophilic attack of the substrate. The MD confirms the stability of the X-ray loop conformations which seem to remain in their X-ray orientations (open or closed); this is stated as true even when acyl-AcpP is excised to generate apo-Fab. I note that it is sometimes confusing when multiple different systems are combined eg. the 12:0 and 16:0 substrate MD data for FabF in Fig S10e could mislead the reader into thinking that loop2 samples both open and closed in the 16:0-AcpP-FabF MD. Should this data be separated?

It does not appear crystal contacts play a role in stabilising the unusual open loop orientations but a comment on this would be welcome. Similarly, it would also be useful to reflect on how the

crosslinker amide NH (H-bond donor) adjacent to the site of attack of Cys163 may alter interactions with the Fab protein, as compared with the S atom (weak H-bond acceptor) that would normally be there. Is there a driving force that could bias the observed flip of the amide such that it orients with the two histidines rather than the expected backbone amides of Cys163 and Phe392?

As a general remark, the only structures presented from the MD simulations are the stylized loop images in Figure 5. It would be helpful to the reader to overlay snapshots, possibly against X-ray structures, to demonstrate more global relaxation of the protein, and assist in commenting on substrate-Fab interactions and substrate conformations (eg. are both chain conformations C16a and C16b sampled in the MD?).

Other comments:

The manuscript is terse, even for a communication. A clear explanation of the crosslinking strategy is required in the text and a Chemdraw figure of the covalent product should be provided in SI to illustrate where in Fig S1b the trapping occurs.

Define in Abstract "AbpP"; define helix I (p6); define "PP" in Fig 1.

Include literature reference when previously solved X-ray structures are mentioned.

Is there a reason why AcpP2 and not AcpP1 is given in Fig 1?

Define clearly 10:0-AcpP.KS complexes etc when first introduced in text (p5)

Star charged residues in Figs S3,S4.

Fig 3a caption needs corrected.

Explain more clearly from outset how Figure S6 shows lack of engagement with oxyanion hole.

It is stated that Fab-AcpP interactions make and break repeatedly over the simulations (para3,p6) – please give timeseries plots in SI as some examples of these.

Refer to the specific panel in Figure S5 when discussing substrate disposition (p7).

Define "d" precisely in text (p7).

What were the reference X-ray distances for the interactions discussed for MD in Fig S7?

The way Fig 3a-c and Fig S5 are referred to is confusing (p7) given they don't contain any MD information.

Define third compound in legend of Fig S1d. Figure S3,S4 – make clear for residue pairs which is from Fab and which from ACP.

“R” appears to have two meanings in the box in Figure S1c.

The sentence “Moreover, extension of the modeled...” needs rewritten for clarity (p9).

Refer to Fig S9,10 in para3,p11

Refer to Fig S11,12 in para1, p12

“phi and psi” not “psi and phi” (eg. p11)

Scheme 1 (p18) does not exist.

“the 1:1” (p18)

“present in” (p18)

It should be stated how independent replicates were initiated (eg. different velocities, initial geometries?).

Berendsen barostat not thermostat (p20)

“such that its edges” (p19)

“covalent bonds involving hydrogen” (p19)

Delete “HF/6-31G*” (p19)

Include as electronic SI the nonstandard residue parameters as AMBER input files.

Delete “also” (para3, p19) as the counterions were part of the salt process.

Reference formatting needs checked (eg. ref 21, 23, 45). Referencing seems to have gone awry eg. refs 68 etc in text don’t exist.

Final page of SI: Be more explicit in the figure as to which parts of the complex are being shown.

Gating Mechanism of β -Ketoacyl-ACP Synthases
NCOMMS-19-14595A
Detailed Response to Reviewer Comments

Item #1, Referee #1: Figures and Figure legends: Almost every figure panel shows structural details in a new view. It needs the careful

reading of the legends to be able to understand which of the many structures is shown or which structures are superimposed. The sequence of figures does moreover not reflect the sequence of references in the text; e.g., see example to Figure 3 and 4 below.

This was addressed by making a clearer introduction of the structure, interface, and associated loop movements. Additionally, this manuscript underwent significant revisions and was modified to focus on gating elements rather than the interface or specific chain length substrate preferences. We believe the manuscript will be easier to follow as figure callouts and general discussion flow in accordance with one another.

Item #2, Referee #1: The timing of introducing information is suboptimal. There is partly too much data introduced simultaneously. The authors should check whether they can build up the surely exciting arguments in a more careful and logical manner; again highlighted below on the basis of one section.

See above comment.

Item #3, Referee #1: Chapter “Conformational heterogeneity in FabF active site loops” starts introducing the C12AcpP-FabF structure, which lacks density of the acyl chain and suffers from poor density of loop 1 owing to conformational heterogeneity. Then it says that “we turned to the 2.30 Å C16AcpP-FabF dataset”. Here the authors set a generic reference to Figure 3,4. There are 16 figure panels in Figure 3 and 4 – are all panels important here? Why not directly starting the chapter with C16AcpP-FabF data?

This was addressed by removing discussion of the C12AcpP-FabF structure from the main text. The sections now begin with C16AcpP-FabF with a more linear analysis of the structural elements.

Item #4, Referee #1: The chapter “Conformational heterogeneity in FabF active site loops” finally closes with MD modeling. The C16AcpP-FabF dataset shows carbonyl group of the fatty acyl chain resides within hydrogen-bonding distance of the two catalytic histidines. MD simulation based on this data obviously increase this distance suggesting that the carbonyl group is not forming hydrogen bonding with either active site histidine so that this state is a pre-transacylation state. At this point, authors refer to Fig. 3a-c. Why referring to X-ray data here? And which C16-chain is shown in Fig. 3c - C16a or C16b?

MD analysis was moved to the second half of the paper to allow a more thorough introduction of the systems simulated and the findings therein. We believe this rectifies issues with referring to X-ray data vs MD simulation data.

Item #4, Referee #1: The Specific reference to panels 3d and 3e (except the generic “Figure 3,4 reference” mentioned above) appear much later in the text, after Fig.4 panels are introduced.

This issue is addressed by removing C12AcpP-FabF from the discussion instead focusing on C16AcpP-FabF earlier in the text.

Item #5, Referee #1: Can the authors comment of the specific value of the structures C12AcpP-FabF and C16AcpP-FabB?

This has been addressed by removing C12AcpP-FabF from general discussion in the main text. Discussion of the C12AcpP-FabF does not provide any additional value to the manuscript.

Item #6, Referee #1: The consensus on the function of F400/F392 in FabF/B (and other KS) is the sensing of the β -carbon modification so that just acyl chains with methylene are accepted for elongation. Data from Zhang et al are in line with such a function (ref 17). Do which extent does the gating mechanism go beyond this concept? Does the gating mechanism allow detecting the correct substrate prior to its elongating into the binding pocket? Or in other words, when does the gating mechanism distinguish between a productive and non-productive interaction (at the “gate open” position)? This is an important point, as the authors also mention in the introduction: “... Therefore, AcpP and its partner enzymes must minimize the number of non-productive PPIs”. But this points remains vague.

We agree with the comments of referee #1 and they have been addressed in part in the “**FabF active site loops adopt a “gate-open” conformation to accommodate acyl cargo**” and “**Sequence conservation of loops 1 and 2 suggests gating as a general feature of KSs**” sections. Additionally, we have included a paragraph that proposes more specific functions of the gate and the role of loop 2 in the discussion section.

Item #7, Referee #1: How well are the key arguments of this paper supported by data: (i) Why does structure C12AcpP-FabF differ from C12AcpP-FabB? Why does C16AcpP-FabB differ from C16AcpP-FabF? Maybe FabF and FabB are more different that anticipated here? (ii) The gating mechanism is built on mainly C12AcpP-FabB. One may also argue that C12AcpP-FabB is simply an artifact/local minimum structure received under the crystallization conditions? Is there more evidence for the gating mechanism? MD data may be supportive here. A better readability of the manuscript may easily clarify these points.

The manuscript underwent significant revisions and was modified to focus on gating elements rather than the interface or specific chain length substrate preferences. Discussion of C12AcpP-FabF and C16AcpP-FabB have been removed from the manuscript. Additionally, assays on FabF gating mutants as well as a more focused computational analysis of these complexes should provide more evidence for the gating mechanism proposed in this work.

Item #8, Referee #1: There are references to recent literature missing. Pertinent research in reference to structures and function of KSs and the engineering of KS domains have been included in the 3rd paragraph of the introduction.

Item #9, Referee #1: The references to Figure panels are inaccurately placed (see also above). This has been addressed in the text

Item #10, Referee #1: Why is the loop with residues range 399-402 numbered with loop 1 and the (N-terminal) loop with residues 263-275 loop 2. This is counter-intuitive.

The loop range is ordered by proximity to the active site instead of the order it appears in primary sequence. Additionally, given that loop 1 is the more conserved aspect of the gate, we place it with higher importance and have labelled it loop 1. We hope that these loops can be given more concrete terms in regard to their function as more work on these systems is completed.

Item #11, Referee #1: Residue Phe392 is incorrectly termed Phe400 in Chapter “Acyl-AcpP-FabB locks into a catalytically competent, “gate-closed”. Please check numbering throughout the manuscript.

This has been addressed in the text.

Item #12, Referee #1: Residues in Figure 6 should not be labeled in FabF residue numbering. It can be misleading when blurring the origin of data?

We respect the concerns of the reviewer here, but the number is left as FabF numbering to present the sequence of events as a general mechanism employed by ketosynthases to process their substrates. We have modified the caption to do our best to be as transparent about this and state that all residues are represented in FabF numbering for ease of understanding their function in the mechanism.

Item #13, Referee #1: Improve figure style: Reduce view. Reduces panel numbers; e.g., by connect panels (e.g. cartoon representations in Figure 5 can be shown as inset). Indicate “dataset” directly in the figure panel. Improve color code (show color code legend). ...

We have reworked a number of figures in the main text. These include adding insets into figure 3 (interface figure) as well as insets in figure 7 (previously figure 5). A color coded legend has been provided in figure 5 (previously figure 4) and a perspective eye is provided to orient the reader for the subsequent loop figure panels.

Item #1, Referee #2: However, my opinion is that the manuscript is not ready for publication. The authors may be reading into their structures too much. While the double-drawbridge gate is an interesting idea, only crystals of the ACP-FabF complexes show the movement of these loops.

We have addressed these concerns by performing a more thorough mutational analysis of residues involved in the proposed gating mechanism. We have included these results as a new section titled “**Mutagenesis studies demonstrate the importance of gating on KS activity** “. These new studies employ the use of crosslinkers with and without associated substrate mimetics. The results demonstrate that gating elements are important for allowing active site access for acyl substrates but not for the entrance of small substrates more akin to malonyl-AcpP, which is in line with our proposed mechanism. Additionally, mutations that we term “pocket block” mutations that block the open position of Phe400 have the lowest crosslinking efficiency with both crosslinkers, demonstrating the importance of loop movements to an open state in order to accept acyl-AcpP substrates. These new assays are coupled by a more developed and focused analysis of our computational data.

Item #2, Referee #2: Despite the authors attempts, the proposed gating mechanism lacks molecular detail. Does FabF use this to select against partially processed acyl chains? How would KS/CLF's use these same motifs?

We agree with referee 2 that these are important considerations and they have been addressed in the following sections, “**Sequence conservation of loops 1 and 2 suggests gating as a general feature of KSs**” and “**FabF active site loops adopt a “gate-open” conformation to accommodate acyl cargo**”. Additionally, there is further discussion of loop 2 function in the discussion section.

Item #3, Referee #2: An active site conformational change does not seem necessary in the function of these enzymes.

These concerns have been addressed in the additional studies and analysis included in the manuscript. See section “**Mutagenesis studies demonstrate the importance of gating on KS activity**” and “**Sequence conservation of loops 1 and 2 suggests gating as a general feature of KSs**”

Additionally, previously published structures of ketosynthases demonstrate that in the apo form the F400 residue blocks access to the active site cysteine residue as well as the fatty acid binding pocket. Our C12AcpP-FabB structure indicates that the oxyanion hole likely engages the fatty acid carbonyl only once it is in the binding pocket. While short chain fatty acids may be able to access this pocket without much movement of loop 1 or F400, longer substrates such as C10 or greater would need significantly more room to access such productive transacylation conformation.

Item #4, Referee #2: The authors conduct molecular dynamics simulations, paying close attention to residues in these loops, but its not clear that they have learned much from these studies, which make up a considerable portion of the text.

This has been addressed by moving MD simulation data and analysis into its own section. This

has allowed us to more clearly describe the simulated systems and use the information from these simulations to support the main thesis of the manuscript. We have focused the analysis only on data that pertains to the function of gates in KSs. This includes a detailed analysis of correlated motions of the loops and a time-resolved analysis of key interloop interactions.

Item #5, Referee #2: My large preference would be to replace this with functional assays of point mutants. For instance, how important is the threonine residue on loop 2? Is there an equivalent residue in FabB's loop 2? Site-directed mutagenesis studies to back up the claimed gating mechanisms are necessary (other ACP/KS interfacial residues could also be tested).

We agree with reviewer 2 that additional mutagenesis studies would be useful to support our conclusions. We have performed such studies and they are enumerated in section “**Mutagenesis studies demonstrate the importance of gating on KS activity**”. These mutations generally focus on residues important for gate function but do also include two interface mutants (K65E and K65A/K69A) that both show a decrease in crosslinking activity.

Item #1, Referee #3: The purpose of the MD simulations within this work should be more clearly stated in the text.

We have expanded and clarified our discussion of the MD simulation work. This include a detailed analysis of correlated motions of the loops and a time-resolved analysis of key interloop interactions. We also provided the data for each system separately as an SI figure.

Item #2, Referee #3: It does not appear crystal contacts play a role in stabilizing the unusual open loop orientations but a comment on this would be welcome.

This comment has been addressed at the end of the section “Changes in the FabF active site accompany substrate binding.”

Item #3, Referee #3: Similarly, it would also be useful to reflect on how the crosslinker amide NH (H-bond donor) adjacent to the site of attack of Cys163 may alter interactions with the Fab protein, as compared with the S atom (weak H-bond acceptor) that would normally be there. Is there a driving force that could bias the observed flip of the amide such that it orients with the two histidines rather than the expected backbone amides of Cys163 and Phe392?

While differences in conformation that result due to replacement of the thioester with an amide cannot be ruled out, we do see that our crosslinker’s carbonyl group is coordinated in the oxyanion hole of FabB in our C12AcpP-FabB structure. Therefore, we do not immediately presume that it would play a role in driving interactions with the histidines in our C16AcpP-FabF structure. We instead think that the differences in the active site architecture, such as the Ile108 residue that blocks access to FabF’s fatty acid binding pocket or the differences in loop 2 between FabF and FabB, may be more determinant in driving the conformation seen in the active site. This is not directly addressed in the text as we have reworked and refocused the paper to address loops 1 and 2 and their role in gating.

Item #4, Referee #3: As a general remark, the only structures presented from the MD simulations are the stylized loop images in Figure 5. It would be helpful to the reader to overlay snapshots, possibly against X-ray structures, to demonstrate more global relaxation of the protein, and assist in commenting on substrate-Fab interactions and substrate conformations (eg. are both chain conformations C16a and C16b sampled in the MD?).

We now show plots of the backbone and sidechain root mean square fluctuation (RMSF) mapped onto the structures of the complex. In addition, we also present RMSD data for both complexes as well as the KS and ACP subunit.

Item #4, Referee #3: The manuscript is terse, even for a communication. A clear explanation of the crosslinking strategy is required in the text and a Chemdraw figure of the covalent product should be provided in SI to illustrate where in Fig S1b the trapping occurs.

In addition to Figure S1, we have added scheme 1 to address this issue. Additionally we have added a maintext figure (figure 2) to address issues with crosslinking probe structure and the one-pot to create crypto-ACPs.

Item #5, Referee #3: Define in Abstract “AbpP”; define helix I (p6); define “PP” in Fig 1. These items have been addressed in the text

Item #6, Referee #3: Include literature reference when previously solved X-ray structures are mentioned.

These items have been addressed in the text

Item #7, Referee #3: Is there a reason why AcpP2 and not AcpP1 is given in Fig 1?

The electrostatic interfaces of both AcpP1 and AcpP2 are equivalent, only one is shown to limit the size and congestion of the figure.

Item #8, Referee #3: Define clearly 10:0-AcpP.KS complexes etc when first introduced in text (p5)

This has been addressed in the text in section “**MD simulations support a gating mechanism involving loops 1 and 2.**”

Item #10, Referee #3: Fig 3a caption needs corrected.

This item has been addressed in the text

Item #11,12,14,16,18,19,20,33 Referee #3: Explain more clearly from outset how Figure S6 shows lack of engagement with oxyanion hole.

The manuscript underwent significant revisions and was modified to focus on gating elements rather than the interface or specific chain length substrate preferences. Due to size constraints and the need to develop the central thesis of the manuscript by including new biochemical and computational analysis content pertaining to this comment has been omitted from the text for brevity.

Items #12,14,16,18,19,20,33 Referee #3: It is stated that Fab-AcpP interactions make and break repeatedly over the simulations (para3,p6) – please give timeseries plots in SI as some examples of these.

(See response above item#11)

Item #13, Referee #3: Define “d” precisely in text (p7).

This item has been addressed in the text

Item #14, Referee #3: What were the reference X-ray distances for the interactions discussed for MD in Fig S7? (See response above item#11)

Item #15, Referee #3: The way Fig 3a-c and Fig S5 are referred to is confusing (p7) given they don't contain any MD information.

This has been addressed by rewriting the manuscript to introduce information in a more linear manner. Additionally, the computational work has been separated into a separate section as to allow a better introduction and discussion of the simulated systems and results.

Item #16, Referee #3: Define third compound in legend of Fig S1d. Figure S3,S4 – make clear for residue pairs which is from Fab and which from ACP. (See response above item#11)

Item #17, Referee #3: “R” appears to have two meanings in the box in Figure S1c.

These are delineated as R^1 and R , where R^1 is in reference to simulated systems and R for the crosslinker is a general R with no specific chain length restrictions in this figures. Crosslinkers used in this manuscript have been clearly defined in main text figure 2.

Item #18, Referee #3: The sentence “Moreover, extension of the modeled...” needs rewritten for clarity (p9). (See response above item#11)

Item #19, Referee #3: Refer to Fig S9,10 in para3,p11(See response above item#11)

Item #20, Referee #3: Refer to Fig S11,12 in para1, p12(See response above item#11)

Item #21, Referee #3: “phi and psi” not “psi and phi” (eg. p11)
This item has been addressed in the text

Item #22, Referee #3: Scheme 1 (p18) does not exist.
This item has been addressed in the text

Item #23, Referee #3: “the 1:1” (p18)
This item has been addressed in the text

Item #24, Referee #3: “present in” (p18)
This item has been addressed in the text

Item #25, Referee #3: It should be stated how independent replicates were initiated (eg. different velocities, initial geometries?).
This item has been addressed in the methods section

Item #26, Referee #3: Berendsen barostat not thermostat (p20)
This item has been addressed in the methods section

Item #27, Referee #3: “such that its edges” (p19)
This item has been addressed in the text

Item #28, Referee #3: “covalent bonds involving hydrogen” (p19)
This item has been addressed in the text

Item #29, Referee #3: Delete “HF/6-31G*” (p19)
This item has been addressed in the text

Item #30, Referee #3: Include as electronic SI the nonstandard residue parameters as AMBER input files.
Parameter files are now provided for the nonstandard residues

Item #32, Referee #3: Reference formatting needs checked (eg. ref 21, 23, 45). Referencing seems to have gone awry eg. refs 68 etc in text don't exist.
Referencing issues have been addressed.

Item #32, Referee #3: Final page of SI: Be more explicit in the figure as to which parts of the complex are being shown. (See response above item#11)

Reviewers' comments:

Reviewer #1 (Remarks to the Author):

There is plenty of information to report in this manuscript. However, while the original version of the manuscript was in parts difficult to read, there is now a very good timing in introducing the wealth of methods, information and figures.

The paper mainly benefits from more clearly separating the molecular dynamics section from Xray crystallographic data interpretation. In the revised version, the Xray section is further more carefully introducing the crystallized complexes. This is all helpful for the reader. The manuscript also benefits from referring to PDB 5KOF that is meanwhile published. The study moreover greatly benefits from the new experimental data on cross-link efficiency read-out by SDS-PAGE (Fig. 5), which targets the role of loop 1 carrying the gate-keeping phenylalanine, as well as loop 2 with its conserved aspartate. While crosslinking data on the loop 1 mutants support the current understanding of the gating role of Phe, mutants of loop 2 show the relevance of Asp for the loading of the substrate. Loop 2 mutations are suggested to support the double drawbridge mechanisms.

This is an elaborate and outstanding study that significantly develops the understanding of the condensation reactions taking place in FASs and PKSs. Publication is clearly recommended.

Comments to consider for a final revision:

-The new biochemical data is interesting and increase the value of the manuscript for the community. Loop 1 mutations are well in line with structural data: F400A improves crosslinking efficiency owing to the removal of the steric barrier and mutation that freeze the closed state decrease efficiency. Data supports the gatekeeping function proposed earlier on the basis of KS-acyl complexes, but is also in line with the new data provided in this manuscript. The D265A and D265N mutations are more difficult to understand. If understood correctly, the argument is that mutations destabilize the open conformations of loops and hinder the loading of longer substrates into the pocket. This is plausible although from the data provided it is not entirely clear why this Asp is more involved in stabilizing the gate-open state than the gate-closed state (small difference in hydrogen bonding pattern – see Fig. 5). The authors should please be clearer in their argumentation and include Asp data again in the discussion section. The sequence alignment (Figure 6) and the computational modeling (although MD data is not giving direct information on an the gate-open conformation) support the role of the conserved Asp. Suggesting a drawbridge model is justified, particularly because phrased carefully: “With this new information, we propose that elongating KSs use a double-drawbridge gating... (page 15).” And: “The proposed gating mechanism coordinated by

loops 1 and 2 is in agreement with our mutagenesis studies, MD simulations, and MSA analysis (again page 15).”

-Page 5 of the pdf: Better say: “AcpP-FabF and AcpP-FabB interfaces can be broken into three regions.”

-Page 11 of the pdf: Better say: “Similar patterns emerge when examining the crosslinking efficiencies of our D265A and D265N mutants of loop 2.” Then it is clear that the paragraph treats the other loop.

-Figure 5 could still be improved for better understanding. It is probably the most important structure figure of the manuscript. Panels a-e are clear (maybe superimpose d and e; or combine b&c and d&e in one panel), but the set f to j is not so simple to read. Think of adding a perspective eye and combining panels as indicated above.

-The discussion on the pong-step should be omitted. It is not clear whether the release of the elongated chain would require more than a slight rotation of the gate-keeping Phe.

Reviewer #2 (Remarks to the Author):

The authors are putting way too much faith in their FabF-C16ACP structure, the only structure that clearly shows the alternate conformation of loops 1 and 2. They entitle the manuscript “Gating Mechanism of beta-Ketoacyl-ACP Synthases” and claim that a double drawbridge mechanism is necessary for an acyl chain to be received and for the extended product to be released. Neither the mutagenesis studies nor the molecular simulation experiments support the authors’ hypotheses. The 16:0 acyl chain, used by the authors, is not efficiently elongated by *E. coli* FabF anyway (16:1 is only a slightly better substrate). They should have tried to capture the complex with a 14:0 substrate, since C14 chains can be efficiently extended by FabF. The structure that probably best represents reality is their FabB-C12ACP structure. In this structure, the carbonyl is in the oxyanion hole and the acyl chain is in its proper acyl binding pocket. As can be seen in this structure, there is no need for loops 1 and 2 to open and close for the chain to access the binding pocket. The only “opening” that can be agreed upon is a slight rotation of F400, but this conformation is also populated in the apo form of these KSs anyhow.

The authors should have stayed focused on what they have learned through their structures (ACP/KS interactions, pantetheinyl arm/KS interactions); however, their recent publication of the 5KOF ACP/KS(FabB) structure in *Nature: Chemical Biology* (“Molecular Basis for Interactions between an

Acyl Carrier Protein and a Ketosynthase”) detracts from this story. Instead, they speculate that gating motions are necessary for KS activity and propose related hypotheses without solid experimental evidence. If this article were accepted, it would adversely affect the fatty acid synthase and polyketide synthase communities. The authors need to recognize that the unnatural crosslinking of proteins can easily result in unphysiological conformations of those proteins being observed by crystallography.

Abstract: “these structures reveal critical conformational states accessed during KS catalysis. KS-catalyzed carbon-carbon bond formation is shown to proceed via a previously unreported gating mechanism during the binding and delivery of acyl-ACPs. Two active site loops undergo large conformational excursions during this dynamic gating mechanism and are likely evolutionarily conserved features in elongating KSs.” – these conformational states were not shown to be critical, and the authors are overinterpreting their data.

p. 4: “Specifically, the KS employs two conserved active site loops that open and close through a double drawbridge-like gating mechanism in order to regulate substrate processing. When open, the drawbridge expands the KS active site in order to accommodate PPant-tethered acyl-AcpP substrates. The structural features underlying this gating machinery are likely conserved in related enzymes. Similar conformational transitions will likely determine substrate selectivity and processing in other FAB and PKB pathways.” – The authors are likely incorrect about the double drawbridge.

p.6: “The carbonyl group of the substrate coordinates to the FabF active site histidine residues, His303 and His340, in a manner similar to the published structure of AcpP-FabB (PDB: 5KOF) (Fig. 4b). – The location of this carbonyl group seems artificial. During catalysis, it should be in the oxyanion hole as observed in their FabB-C12ACP structure.”

p. 10 “Therefore, these iterative type II KSs need to disfavor gate opening to prevent premature product dissociation. In actKS, this is likely realized by destabilizing the open conformation of loops 1 and 2 through the substitution of the negatively charged Asp265 residue of loop 2 with an Asn residue (Fig. 5a,b).” – This is speculation on top of speculation. It also seems the authors are now arguing against the double drawbridge in highly related KSs.

p. 10: “less efficiently with longer C8 and C12 chain length crosslinkers, 2b and 1a, which require transitions between gate conformations to access the KS active site.” and p. 16: “Finally, loops 1 and 2 return to an open conformation to enable product dissociation.” – there are

no compelling data indicating that transitions beyond the slight rotation of the F400 side-chain are required.

Reviewer #3 (Remarks to the Author):

The authors have satisfactorily addressed my comments. From my perspective, no further revision required.

Gating Mechanism of β -Ketoacyl-ACP Synthases
NCOMMS-19-14595A rev. 2
Detailed Response to Reviewer Comments

Reviewer #1 (Remarks to the Author):

There is plenty of information to report in this manuscript. However, while the original version of the manuscript was in parts difficult to read, there is now a very good timing in introducing the wealth of methods, information and figures.

The paper mainly benefits from more clearly separating the molecular dynamics section from Xray crystallographic data interpretation. In the revised version, the Xray section is furthermore carefully introducing the crystallized complexes. This is all helpful for the reader. The manuscript also benefits from referring to PDB 5KOF that is meanwhile published. The study moreover greatly benefits from the new experimental data on cross-link efficiency read-out by SDS-PAGE (Fig. 5), which targets the role of loop 1 carrying the gate-keeping phenylalanine, as well as loop 2 with its conserved aspartate. While crosslinking data on the loop 1 mutants support the current understanding of the gating role of Phe, mutants of loop 2 show the relevance of Asp for the loading of the substrate. Loop 2 mutations are suggested to support the double drawbridge mechanisms.

This is an elaborate and outstanding study that significantly develops the understanding of the condensation reactions taking place in FASs and PKSs. Publication is clearly recommended.

Comments to consider for a final revision:

The new biochemical data is interesting and increase the value of the manuscript for the community. Loop 1 mutations are well in line with structural data: F400A improves crosslinking efficiency owing to the removal of the steric barrier and mutation that freeze the closed state decrease efficiency. Data supports the gatekeeping function proposed earlier on the basis of KS-acyl complexes but is also in line with the new data provided in this manuscript. The D265A and D265N mutations are more difficult to understand. If understood correctly, the argument is that mutations destabilize the open conformations of loops and hinder the loading of longer substrates into the pocket. **This is plausible although from the data provided it is not entirely clear why this Asp is more involved in stabilizing the gate-open state than the gate-closed state (small difference in hydrogen bonding pattern – see Fig. 5). The authors should please be clearer in their argumentation and include Asp data again in the discussion section.** The sequence alignment (Figure 6) and the computational modeling (although MD data is not giving direct information on an the gate-open conformation) support the role of the conserved Asp. Suggesting a drawbridge model is justified, particularly because phrased carefully: “With this new information, we propose that elongating KSs use a double-drawbridge gating... (page 15).” And: “The proposed gating mechanism coordinated by loops 1 and 2 is in agreement with our mutagenesis studies, MD simulations, and MSA analysis (again page 15).”

This is plausible although from the data provided it is not entirely clear why this Asp is more involved in stabilizing the gate-open state than the gate-closed state (small difference in hydrogen bonding pattern – see Fig. 5). The authors should please be clearer in their argumentation and include Asp data again in the discussion section.

We recognize the concerns of the reviewer and have clarified our argument in the manuscript. The Asp265 residue in question is a charged residue, as determined by PROPKA3.0, a pKa analysis software. Therefore, the mutation of Asp265 to an asparagine residue transforms charged interactions between the two loops into polar interactions. There is a significant difference in the noncovalent interaction energies of ion•dipole and dipole•dipole interactions. It is for this reason that we state that mutating the Asp265 to either Ala or Asn will disfavor the gate open conformation as the charged Asp265 would more readily stabilize this state. This is now explicitly stated in the manuscript (pg.11, ¶3)

-Page 5 of the pdf: Better say: “AcpP-FabF and AcpP-FabB interfaces can be broken into three regions.”

We have modified the text as suggested.

-Page 11 of the pdf: Better say: “Similar patterns emerge when examining the crosslinking efficiencies of our D265A and D265N mutants of loop 2.” Then it is clear that the paragraph treats the other loop.

We have modified the text as suggested.

-Figure 5 could still be improved for better understanding. It is probably the most important structure figure of the manuscript. Panels a-e are clear (maybe superimpose d and e; or combine b&c and d&e in one panel), but the set f to j is not so simple to read. Think of adding a perspective eye and combining panels as indicated above.

We have combined panels and added perspective eyes as requested.

-The discussion on the pong-step should be omitted. It is not clear whether the release of the elongated chain would require more than a slight rotation of the gate-keeping Phe.

The discussion of the pong step has been removed.

Reviewer #2 (Remarks to the Author):

The authors are putting way too much faith in their FabF-C16ACP structure, the only structure that clearly shows the alternate conformation of loops 1 and 2. They entitle the manuscript “Gating Mechanism of beta-Ketoacyl-ACP Synthases” and claim that a double drawbridge mechanism is necessary for an acyl chain to be received and for the extended product to be released. **Neither the mutagenesis studies nor the molecular simulation experiments support the authors’ hypotheses. The 16:0 acyl chain, used by the authors, is not efficiently elongated by E. coli FabF anyway (16:1 is only a slightly better substrate). They should have tried to capture the complex with a 14:0 substrate, since C14 chains can be efficiently extended by FabF. The structure that probably best represents reality is their FabB-C12ACP structure. In this structure, the carbonyl is in the oxyanion hole and the acyl chain is in its proper acyl binding pocket. As can be seen in this structure, there is no need for loops 1 and 2 to open and close for the chain to access the binding pocket. The only “opening” that can be agreed upon is a slight rotation of F400, but this conformation is also populated in the apo form of these KSs anyhow. (...)**

The highlighted statements are parsed below.

Neither the mutagenesis studies nor the molecular simulation experiments support the authors’ hypotheses.

- We believe that our interpretation of the mutagenesis studies and MD simulation experiments support the hypothesis of a previously unidentified gating mechanism, and this is now fully described in the text. Given that the additional mutagenesis data was performed at the request of this Reviewer, it is not clear what part of our analysis troubles the Reviewer.
- The mutagenesis studies were performed to test key structural and dynamic aspects of these gating loops to show that access to the active site for acyl-AcpP substrates is hindered when the dynamics of the loops are perturbed by site directed mutations. The assay results demonstrate a substantial decrease in crosslinking activity for probes bearing a long-chain acyl substrate mimetic compared to probes without said long-chain acyl substrate mimetic when testing gating impaired FabF mutants. These results support our hypothesis that larger substrates require loop reorganization to facilitate substrate binding and acyl chain transfer. Our assay results are described in greater detail on page 10 of the manuscript.
- MD simulations demonstrate that the open and closed conformations of loops 1 and 2 are stable. This stabilization indicates that these loop conformations are not artifactual. Furthermore, rigorous simulations also support the role of D265 in forming intra- and inter-loop interactions to stabilize the different conformational states. These conclusions are supported by the additional mutagenesis studies and the conserved nature of the residues that form this network of interactions of loops 1 and 2.

The 16:0 acyl chain, used by the authors, is not efficiently elongated by E. coli FabF anyway (16:1 is only a slightly better substrate). They should have tried to capture the complex with a 14:0 substrate, since C14 chains can be efficiently extended by FabF.

- During Reviewer 2's initial review of the manuscript they stated "The authors may be reading into their structures too much. While the double-drawbridge gate is an interesting idea, only crystals of the ACP-FabF complexes show the movement of these loops." Our C16AcpP-FabF structure and respective analysis was present and central to our model in the initial revision of the paper. We are perplexed that these issues are being raised at this point in the review process.
- Reviewer 2 suggests that we should have tried to capture the complex with a substrate that FabF would extend, and we have done this, namely our C12AcpP-FabF structure that we report in this publication. Please see Supplementary Note 2 for further details. This discussion was removed from the main text discussion at the request of reviewer 1 after the initial review process. Additionally, 12:0-AcpP is a better substrate than 14:0-AcpP. See following for further details:
Edwards, P.; Nelson, J. S.; Metz, J. G.; Dehesh, K. *FEBS Lett.* **1997**, *402*, 62-66.
- Moreover, a 16-carbon length species is found in the active sites of these enzymes, namely the elongation product of 14:0-AcpP. Perhaps, this C16AcpP-FabF structure represents something similar to a "product" bound state. This was our main reasoning for suggesting that the gate may not only facilitate substrate binding but also induce product release.

The structure that probably best represents reality is their FabB-C12ACP structure. In this structure, the carbonyl is in the oxyanion hole and the acyl chain is in its proper acyl binding pocket. As can be seen in this structure, there is no need for loops 1 and 2 to open and close for the chain to access the binding pocket. The only "opening" that can be agreed upon is a slight rotation of F400, but this conformation is also populated in the apo form of these KSs anyhow.

- These two synthases are not identical in sequence or structure. FabB is known to produce unsaturated fatty acids and FabF is known to participate in homeoviscous adaption via the increased production of C18:1 in response to decreases in temperature (Supplementary Note S4). A reasonable argument can be made that trapping intermediate states of these complexes may very well be dictated by these differences in sequence, structure, and activity.
- The Reviewer states that the C12AcpP-FabB structure is the only relevant structure, as the substrate is bound and coordinated in the oxyanion hole. However, the binding pockets of FabF and FabB differ from one another. The FabB binding pocket is distinct and has no Ile108 gating residue, allowing easier access for substrates. Therefore, it is likely that the C12 substrate is more easily accommodated in the active site, as opposed to that of FabF which contains a I108 residue that impedes the binding pocket at the C10-C12 range. This topic is discussed at length in Supplemental Note S4.
- We would like to highlight that x-ray crystallography typically reveals a single (or few) discrete low-energy states. Therefore, it cannot be reasonably concluded that a dynamic process such as loop motions do not preclude substrate binding and positioning on the basis of a (static) x-ray structure of a substrate-bound complex.

- This comment states that the only “opening” that can be agreed upon is a slight rotation of Phe400. While we agree that the literature is quite sound in regard to a slight rotation of Phe400 to allow malonyl-ACP binding to the acyl-KS intermediate, the mutagenesis assay results show that larger loop motions are required for allowing active site access for longer chain acyl-ACP substrates.

The authors should have stayed focused on what they have learned through their structures (ACP/KS interactions, pantetheinyl arm/KS interactions); however, their recent publication of the 5KOF ACP/KS(FabB) structure in Nature: Chemical Biology (“Molecular Basis for Interactions between an Acyl Carrier Protein and a Ketosynthase”) detracts from this story. Instead, they speculate that gating motions are necessary for KS activity and propose related hypotheses without solid experimental evidence. **If this article were accepted, it would adversely affect the fatty acid synthase and polyketide synthase communities.** The authors need to recognize that the unnatural crosslinking of proteins can easily result in unphysiological conformations of those proteins being observed by crystallography.

The highlighted statements are parsed below.

The authors should have stayed focused on what they have learned through their structures (ACP/KS interactions, pantetheinyl arm/KS interactions); however, their recent publication of the 5KOF ACP/KS(FabB) structure in Nature: Chemical Biology (“Molecular Basis for Interactions between an Acyl Carrier Protein and a Ketosynthase”) detracts from this story.

- This manuscript has always emphasized the structural changes in the enzyme active site that are revealed in our crosslinked structures. In fact, this has not changed since the manuscript’s initial submission, except for the addition of the description of interface mutants as requested by this Reviewer in the first round of review. In fact, we also provide a time-resolved analysis of the AcpP•KS interfaces using rigorous MD simulations. This analysis is provided in the Supplementary Note S5 and Supplementary Figures S5, S32, S33.
- While a comparison of the interfaces of FabF and FabB are briefly discussed in the manuscript, the more significant findings are related to active site loop conformations, as they have broader implications for the function and activities of both FAS and PKS enzymes. We believe that the story as we have presented it is more in line with the high impact research and new findings published in *Nature Communications*.

Instead, they speculate that gating motions are necessary for KS activity and propose related hypotheses without solid experimental evidence.

- While we welcome criticism to increase the value and impact of our work, the Reviewer does not provide constructive commentary as to why our work is speculation or that there is no solid experimental evidence to support our claims. This manuscript discusses 4 crosslinked crystal structures, a mutagenesis study, extensive MD simulations, and the notable conservation of loop elements as seen by MSA analysis. By no means are we

suggesting that our results are absolute; rather, we are proposing a model that is in line with our crystallographic models and supporting experimental and computational data.

If this article were accepted, it would adversely affect the fatty acid synthase and polyketide synthase communities. The authors need to recognize that the unnatural crosslinking of proteins can easily result in unphysiological conformations of those proteins being observed by crystallography.

- While we acknowledge the skepticism of Reviewer 2, we can say that this manuscript does not contradict anything currently known about ketosynthases in FAS or PKS pathways. Rather, it extends our understanding of these complex enzymes. This work is supported by structural data, mutagenesis data, MD simulations, and sequence analysis. We feel that these current results support our hypothesis of a gating mechanism. Of course, we are open to constructive criticism about these results and have demonstrated a willingness to carry out additional experiments to further support our working hypothesis.
- This Reviewer did not raise any issues regarding crosslinking artifacts in the first review process and instead suggested mutagenesis studies. Now that these experiments have been carried out, the Reviewer raises new concerns with our x-ray structures. While we acknowledge these concerns, we have provided additional data and analyses to support our model at the request of this Reviewer. Without clear articulation of why these results are invalid or how they could be improved, we suggest that the work is ready to be tested by others.
- Until recently, structures of acyl-ACP-KS complexes have been elusive due to the transient association of AcpP with its enzymatic partners. These enzymes operate on acyl-ACP substrates; therefore, analysis of these complexes is crucial to the study and future engineering of these enzymes. Our results provide some of the first analyses of a substrate-bearing ACP bound in a KS active site. The only way this could be achieved, that we know of, is through crosslinking with probes we have developed. While we admit that no chemical biology tool is an exact replacement for a natural substrate, these probes represent the closest proxies of native substrates that can be used to trap the ACP in association with partner KSs. Further, these tools have significant precedent in the literature by our group and multiple others who have adopted the technology.

Abstract: “these structures reveal *critical* conformational states accessed during KS catalysis. KS-catalyzed carbon-carbon bond formation is ~~shown to~~ proposed to proceed via a previously unreported gating mechanism during the binding and delivery of acyl-ACPs. Two active site loops undergo large conformational excursions during this dynamic gating mechanism and are likely evolutionarily conserved features in elongating KSs.” – these conformational states were not shown to be critical, and the authors are overinterpreting their data.

- We have removed the word “critical” from the quoted region of the abstract to emphasize that our proposed mechanism, while plausible on the basis of our data, is a proposal.
- We have removed “shown to” and replaced it with proposed to in order to emphasize that this is a newly proposed model based on the data presented manuscript provides.

p. 4: “Specifically, the KS employs two conserved active site loops that open and close through a double drawbridge-like gating mechanism in order to regulate substrate

processing. When open, the drawbridge expands the KS active site in order to accommodate PPant-tethered acyl-AcpP substrates. The structural features underlying this gating machinery are likely conserved in related enzymes. Similar conformational transitions will likely determine substrate selectivity and processing in other FAB and PKB pathways.” – The authors are likely incorrect about the double drawbridge.

- We have modified the manuscript such that a detailed description of the gating mechanism no longer precedes a discussion of our experimental findings.

p.6: “The carbonyl group of the substrate coordinates to the FabF active site histidine residues, His303 and His340, in a manner similar to the published structure of AcpP-FabB (PDB: 5KOF) (Fig. 4b). – The location of this carbonyl group seems artifactual. During catalysis, it should be in the oxyanion hole as observed in their FabB-C12ACP structure.”

- The quoted section of the text states that both the previously published 5KOF structure and our current C16AcpP-FabF structure both coordinate to the histidines. Is the Reviewer suggesting that both structures are artifactual?
- Our central argument has not changed significantly from the original submission of the manuscript. Namely, we posit that the C16AcpP-FabF structure represents a trapped intermediate conformation, either before acyl chain transfer or during substrate dissociation, that reveals a new gating mechanism. We argue that the most recent mutagenesis and MD studies support this hypothesis.
- FabF and FabB have different active site elements that cannot be ignored. Namely FabB does not contain a gating residue in its fatty acyl binding pocket, while FabF has an Ile108 positioned 10-12 carbon lengths from the active site Cys. Additionally, loop 2 from these two synthases are conserved within, but not between, the condensing enzyme families. Arguably, these differences in sequence, structure, and function may lead to the stabilization of the different conformations seen in these synthases. This is discussed in detail in our Supplementary Note S4.
- Our structures have not changed from the original submission. Without further clarification, we are concerned that these issues were not raised during the first round of review, especially in light of the time and effort expended to perform the additional studies requested.
- On the basis of available simulation data, the substrate carbonyl does not migrate to the oxyanion hole. Based upon our proposed mechanism, loop reorganization would have to precede substrate reorganization, which operate on a longer μ s timescale. This is line with the reported FabF and FabB turnover rates (2-3 min⁻¹). For further details, see:
Borgaro, J. G.; Chang, A.; Machutta, C. A.; Zhang, X.; Tonge, P. J. *Biochemistry* **2011**, *50*, 10678-10686.

p. 10 “Therefore, these iterative type II KSs need to disfavor gate opening to prevent premature product dissociation. In actKS, this is likely realized by destabilizing the open conformation of loops 1 and 2 through the substitution of the negatively charged Asp265 residue of loop 2 with an Asn residue (Fig. 5a,b).” – This is speculation on top of speculation. It also seems the authors are now arguing against the double drawbridge in highly related KSs.

- In the first round of review, this Reviewer requested that we provide a broader discussion of loops 1 and 2 in related synthases, specifically those of type II PKS systems. From the prior review, “How would KS/CLF's use these same motifs?” We addressed this question in the revised manuscript with the quoted text above. These statements do not contradict our proposed mechanism, but rather are consistent with that mechanism. To address this concern, we have further clarified this discussion (pg 10, p1) to better explain how differences in loop composition may alter loop interactions.
- For clarity here, our main point is that iterative type II PKS KSs, such as actinorhodin KS/CLF, do not accept acyl-ACPs but instead use malonyl-ACP for iterative condensation reactions. This would mean that the gate would not need to open as only a slight rotation of F400 is necessary in this specific case. In fact, opening of the gate could potentially be detrimental to enzyme function as this may allow premature dissociation of an intermediate poly- β -keto-ACP instead of transferring the polyketide back to the KS cysteine residue. We feel that the modified manuscript now more accurately reflects these statements.

p. 10: “less efficiently with longer C8 and C12 chain length crosslinkers, 2b and 1a, which require transitions between gate conformations to access the KS active site.” and p. 16: “Finally, loops 1 and 2 return to an open conformation to enable product dissociation.” – there are no compelling data indicating that transitions beyond the slight rotation of the F400 side-chain are required.

- The two quoted sentences are derived from different portions of the manuscript making it difficult to understand the reviewer’s concerns. To address this comment, and that of reviewer 1, we have removed the quoted sentence “Finally, loops 1 and 2 return to an open conformation to enable product dissociation.” from the manuscript.

Reviewer #3 (Remarks to the Author):

The authors have satisfactorily addressed my comments. From my perspective, no further revision required.